# Discrete Diffusion VLA: Bringing Discrete Diffusion to Action Decoding in Vision-Language-Action Policies

## Abstract

Vision–Language–Action (VLA) models adapt large vision–language backbones to map images and instructions into robot actions. However, prevailing VLAs either generate actions autoregressively in a fixed left-to-right order or attach separate MLP or diffusion heads outside the backbone, leading to fragmented information pathways and specialized training requirements that hinder a unified, scalable architecture. We present Discrete Diffusion VLA, a unified-transformer policy that models discretized action chunks with discrete diffusion. The design retains diffusion's progressive refinement paradigm while remaining natively compatible with the discrete token interface of VLMs. Our method achieves an adaptive decoding order that resolves easy action elements before harder ones and uses secondary re-masking to revisit uncertain predictions across refinement rounds, which improves consistency and enables robust error correction. This unified decoder preserves pretrained vision-language priors, supports parallel decoding, breaks the autoregressive bottleneck, and reduces the number of function evaluations. Discrete Diffusion VLA achieves 96.3% avg. success rates on LIBERO, 71.2% visual matching on SimplerEnv-Fractal and 54.2% overall on SimplerEnv–Bridge, improving over autoregressive, MLP decoder and continuous diffusion baselines. These findings indicate that discrete-diffusion VLA supports precise action modeling and consistent training, laying groundwork for scaling VLA to larger models and datasets.

## 1 Introduction

Vision-Language-Action (VLA) models enable robots to interpret visual and linguistic inputs and execute corresponding action sequences. Modern VLA frameworks typically adapt a large pretrained vision-language model (VLM) by adding an action-generation head that outputs motor commands (either continuous trajectories or discrete tokens). Current approaches fall into two paradigms: (1) an autoregressive (AR) approach, inspired by GPT-style transformers, that predicts discretized action tokens sequentially within the transformer (*e.g.* OpenVLA (Kim et al., 2024), $\pi_0$-FAST (Pertsch et al., 2025)); and (2) a separate action decoder that employs MLP or continuous diffusion to map VLM output latent tokens to executable controls (*e.g.*, $\pi_0$ (Black et al., 2024) and SmolVLA (Shukor et al., 2025)). Continuous diffusion can model sophisticated multimodal actions better than AR but remains decoupled from the VLM backbone. Some integration efforts (*e.g.*, Transfusion (Zhou et al., 2024) in $\pi_0$) still rely on diffusion-specific training and iterative sampling, lacking a truly unified structure which is consistent with VLM part.

Drawing on recent advances of discrete diffusion and discrete flow-matching for language and multimodal generation (Nie et al., 2025a; Shi et al., 2024b; Gat et al., 2024; Kim et al., 2025a; Yang et al., 2025a), we introduce Discrete Diffusion VLA, the first VLA framework to unify vision, language, and action in a single transformer, maintaining strong VLM priors and achieving precise action modeling. In Discrete Diffusion VLA, each action dimension is first discretized into tokens via binning scheme and then grouped into fixed-length chunks. The fixed-length token generation is exactly suitable for discrete diffusion models. During training, we mask a subset of tokens in the action chunk and train the transformer to predict them from the context of unmasked tokens across all modalities. At inference, we start with all action tokens masked and iteratively predict

Figure 1: **Paradigm comparison.** Continuous diffusion over action chunks (left) versus discrete token decoders: AR (sequential), BERT-style (parallel), and our discrete diffusion with re-masking.

them and re-mask low-confident ones until convergence, in line with the "first-easy, then-hard" philosophy. Moreover, we employ a secondary re-masking technique to guarantee the consistency across different denoising steps, yielding flexible parallel decoding and robust error correction.

For robotics manipulation tasks, our Discrete Diffusion VLA keeps action generation inside a unified transformer with the same training objective (*i.e.*, cross-entropy loss) as VLM, different from separate action decoder methods. This preserves much of the backbone's pretrained vision and language capability (analogous to extending a LLM to new languages), while offering a potential path to inherit from unified transformer's scaling behavior, paving the way for future large-scale VLA research. On the other hand, Discrete Diffusion VLA breaks AR model's left-to-right bottleneck. Action chunks are adaptively decoded in parallel over a small number of steps, and unconfident tokens can be revisited via iterative re-masking, leveraging full cross-modal context (including actions itself) for refinement.

We evaluate Discrete Diffusion VLA across multiple robots and tasks: (1) a Franka Panda arm on LIBERO (Liu et al., 2023), (2) a Google Robot on SimplerEnv–Fractal (Li et al., 2025), and (3) a WidowX Arm on SimplerEnv–Bridge (Li et al., 2025). Using only RGB inputs, language and end-effector positions (no depth and affordances), Discrete Diffusion VLA achieves 96.3% average success rates on LIBERO ( +0.9% vs. OpenVLA-OFT (Discrete) ), 71.2% visual matching with 64.1% overall on SimplerEnv–Fractal, and 54.2% overall on SimplerEnv–Bridge (+14.7% over $\pi_0$ and +6.4% over $\pi_0$-FAST). Our model outperforms all of AR, MLP decoder and diffusion baselines across tasks while using fewer number of function evaluations (NFEs) than AR. Ablations confirm the benefits of our adaptive decoding strategy.

In summary, our contributions are threefold: 1) We introduce the first discrete diffusion VLA, unifying action generation with vision–language in one transformer while maintaining strong performance. 2) We develop an adaptive decoding strategy with iterative re-masking, enabling parallel action-token decoding and error correction, improving accuracy with the unified architecture. 3) We validate Discrete Diffusion VLA on Franka Panda, Google Robot and WidowX, achieving 96.3% avg. SR on LIBERO, 64.1% and 54.2% overall on SimplerEnv–Fractal and –Bridge, consistently outperforming AR, MLP decoder and continuous diffusion baselines (*e.g.*, $\pi_0$ and $\pi_0$-FAST).

## 2 RELATED WORKS

### 2.1 VISION-LANGUAGE-ACTION MODELS

Early VLA systems took a two-part form: RT-1 and RT-2 first let the VLM produce latent tokens, then a separate MLP action decoder maps them to discretized controls in a single shot (Brohan et al., 2022; Zitkovich et al., 2023). Subsequent work shifted to AR token policies and scaled backbones and vision encoders for general manipulation (*e.g.* OpenVLA) (Kim et al., 2024; Touvron et al., 2023b; Oquab et al., 2024a; Zhai et al., 2023), while learning latent action shows better versatility (*e.g.* LAPA (Ye et al., 2024)). For high-frequency control and fast adaptation, token compression and action chunking are introduced and improve efficiency (*e.g.* $\pi_0$-FAST and OpenVLA-OFT) (Pertsch et al., 2025; Kim et al., 2025b). In parallel, diffusion and flow matching action heads model continuous trajectories (Janner et al., 2022; Chi et al., 2023; Liang et al., 2023; Liu et al., 2024), spanning lightweight transformers (Li et al., 2024a), and hierarchical designs (*e.g.* $\pi_0/\pi_{0.5}$) (Black et al., 2024; Intelligence et al., 2025; Bu et al., 2025b;a; Wen et al., 2025; Zhong et al., 2025; Liu et al., 2025), typically via a separate denoising loop conditioned on language and vision tokens.

In contrast, we perform discrete diffusion inside a single transformer over tokenized action chunks, enabling parallel, revisable decoding via iterative re-masking in a few steps while preserving the VLM's emergent multimodal capabilities.

## 2.2 DISCRETE DIFFUSION MODELS

Discrete diffusion has recently achieved strong results on discrete data, notably tokenized images and natural language. Foundational work like D3PM (Austin et al., 2021) formalizes discrete diffusion as a Markov chain. Different from continuous diffusion models, it factorizes across positions, so each token (represented as a one-hot vector) is independently corrupted into a categorical distribution. VQ-Diffusion (Gu et al., 2022) and MaskGIT (Chang et al., 2022a) build on this, achieving high-fidelity image generation with transformers that iteratively predict masked/corrupted image tokens in a non-autoregressive manner. In language, Diffusion-BERT (He et al., 2022) and subsequent Masked Diffusion Models (Shi et al., 2024c; Zheng et al., 2024) demonstrate the viability of this approach and more recent LLaDA (Nie et al., 2025b) and DiffuLLaMA (Gong et al., 2024) scale it to 7B LMs competitive with AR baselines. MMaDA (Yang et al., 2025a) further develops this direction and shows the unified discrete diffusion model can jointly generate text and images.

Our work extends this line to the *action* modality: we perform discrete diffusion over tokenized action chunks, preserving language capabilities and VLM synergy, yielding competitive state-of-the-art VLA performance and laying groundwork toward unified, scalable, multi-modal foundation models spanning vision, language, and actions.

## 3 DISCRETE DIFFUSION VISION-LANGUAGE-ACTION MODEL

### 3.1 OVERVIEW

Figure 2 outlines our Discrete Diffusion VLA. We cast action decoding as discrete diffusion via masked-token denoising *inside the same transformer* that encodes vision and language. Given image observations (single- or multi-view) and a language instruction, each continuous control dimension is discretized into tokens and packed into a fixed-length future action chunk. A single transformer attends to frozen visual features and pretrained-LM text embeddings while progressively unmasking action tokens according to a diffusion schedule, so perception, instruction grounding, and action denoising are executed within one unified model.

Section 3.2 formalizes discrete diffusion over action tokens; Section 3.3 presents the unified transformer architecture; Section 3.4 outlines the overall algorithmic pipeline; and Section 3.5 details inference, including the adaptive decoding mechanism and secondary re-masking for consistency.

### 3.2 DISCRETE DIFFUSION OVER ACTION TOKENS

Let a single action chunk be a length-$L$ token sequence $\mathbf{a}_0 = (a_{0,1}, \ldots, a_{0,L})$, where each $a_{0,i} \in \{1, \ldots, K\}$ is obtained by binning continuous controls (position, orientation, *etc.*) following prior methods (Brohan et al., 2022; Kim et al., 2024). We augment the action vocabulary with a special mask token M (*i.e.*, [MASK]), yielding $V = K+1$ symbols and one-hot basis $\{\mathbf{e}_1, \ldots, \mathbf{e}_K, \mathbf{e}_M\}$.

The **forward** (noising) process of discrete diffusion is a Markov chain $\{\mathbf{a}_t\}_{t=0}^T$ with per-step transition matrices $\mathbf{Q}_t \in \mathbb{R}^{V \times V}$ that independently map each token to M with probability $\beta_t$ and keep it unchanged with probability $1-\beta_t$. Formally, for any one-hot vector $\mathbf{e}_{a_{t,i}}$ of token $a_{t,i}$,

$$\mathbf{Q}_t \, \mathbf{e}_{a_{t,i}} = (1-\beta_t) \, \mathbf{e}_{a_{t,i}} + \beta_t \, \mathbf{e}_M. \tag{1}$$

Composing transition matrices yields $\bar{\mathbf{Q}}_t = \mathbf{Q}_t \cdots \mathbf{Q}_1$, and the corrupted distribution at time $t$ factorizes across positions with $L$ the length of action chunk tokens.

$$q(\mathbf{a}_t \mid \mathbf{a}_0) = \prod_{i=1}^{L} \text{Categorical}\big(a_{t,i} \mid \bar{\mathbf{Q}}_t \, \mathbf{e}_{a_{0,i}}\big), \tag{2}$$

The **reverse** (denoising) process defines conditionals $p_\theta(\mathbf{a}_{t-1} \mid \mathbf{a}_t, \mathbf{c})$ under multimodal (*i.e.*, vision and language) context $\mathbf{c}$. By Bayes' rule, for each position $i$, we have

$$p_\theta(a_{t-1,i} \mid a_{t,i}, \mathbf{c}) \, \propto \, q(a_{t,i} \mid a_{t-1,i}) \, p_\theta(a_{t-1,i} \mid \mathbf{c}), \tag{3}$$

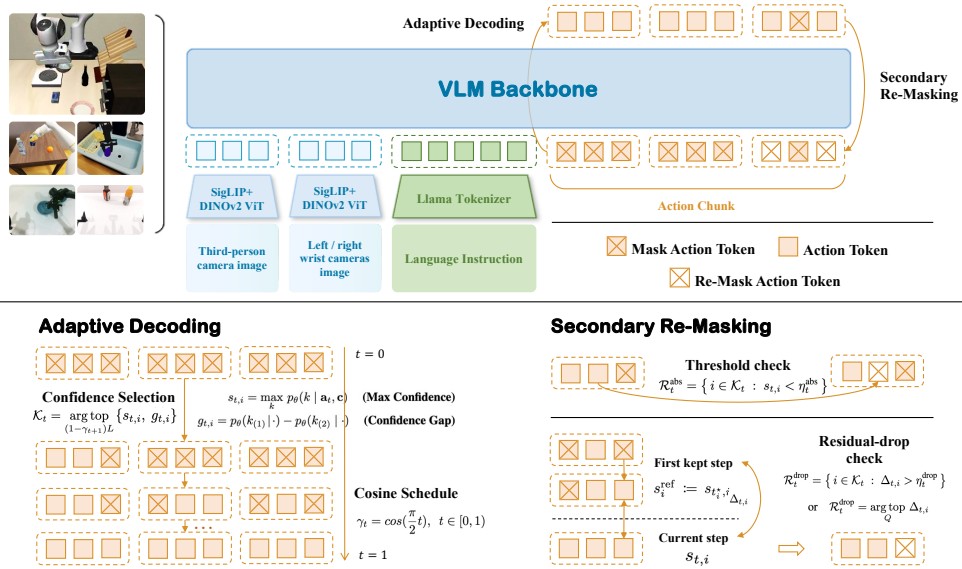

Figure 2: **Discrete Diffusion VLA architecture.** A single-transformer VLM backbone encodes multi-view RGB (SigLIP+DINOv2 ViTs) and a tokenized instruction, and decodes discrete action chunks via diffusion-style iterative refinement. *Adaptive Decoding* (bottom left) keeps high-confidence tokens each round and anneals the keep ratio with a cosine schedule for easy-first parallel refinement. *Secondary Re-Masking* (bottom right) uses threshold and residual-drop checks to re-mask uncertain tokens, enforcing cross-step consistency and robust error correction.

which, under the masking corruption $q$ in Eq. 2, reduces to

$$p_\theta(a_{t-1,i} \mid a_{t,i}, \mathbf{c}) = \begin{cases} \delta(a_{t-1,i} = a_{t,i}), & a_{t,i} \neq \mathrm{M}, \\ \mathrm{Categorical}(a_{t-1,i} \mid \pi_\theta(i \mid \mathbf{c})), & a_{t,i} = \mathrm{M}, \end{cases} \tag{4}$$

where $\pi_\theta(i \mid \mathbf{c})$ is the model's predictive distribution. Thus, at each step, Discrete Diffusion VLA recovers only a subset of masked positions and leaves the rest masked, moving from higher to lower mask ratios until reaching $\mathbf{a}_0$.

In implementation, we follow mask diffusion formulations and collapse the multi-step chain into a single masked-token prediction objective. We draw a mask ratio $\gamma_t \in (0, 1]$ that emulates diffusion time $t$, replace the selected action positions by special token [MASK] to obtain $\tilde{\mathbf{a}}_t$, and train a transformer $f_\theta$ to predict the original tokens with cross-entropy on masked indices:

$$\mathcal{L}_{\mathrm{CE}}(\theta) = -\sum_{i \in \mathcal{M}_{\gamma_t}} \log p_\theta(a_{0,i} \mid \tilde{\mathbf{a}}_t, \mathbf{c}), \qquad p_\theta(\cdot) = \mathrm{softmax}(W f_\theta(\tilde{\mathbf{a}}_t, \mathbf{c})), \tag{5}$$

where $\mathcal{M}_{\gamma_t}$ is the masked set and $W$ projects hidden states of action positions to the $K$-way action local vocabulary. This objective preserves diffusion's corruption–denoising spirit while using a simple maximum-likelihood surrogate. As shown in recent analyses (Shi et al., 2024a; Kim et al., 2025a), such losses upper-bound the negative log-likelihood under appropriate schedules.

Discrete Diffusion VLA accepts increased training-time complexity (*i.e.* solving exponentially many infilling tasks) to gain *arbitrary-order* decoding at test time, selecting the inference order adaptively by confidence or confidence gap, different from BERT-style parallel decoding which uses a fixed, small mask ratio in a single pass and lacks a principled generative reverse chain.

### 3.3 OUR UNIFIED VLA ARCHITECTURE

**Base architecture.** We build upon OpenVLA (Kim et al., 2024) architecture consisting of a Prismatic-7B VLM (Karamcheti et al., 2024) with SigLIP+DINOv2 (Zhai et al., 2023; Oquab et al., 2024b) visual encoders, a projector, and a Llama 2 language model (LM) backbone (Touvron et al., 2023a). Unlike the original autoregressive action head, Discrete Diffusion VLA modifies the causal-attention backbone into a bi-direction transformer that embeds discrete diffusion over actions inside the same VLM backbone, yielding a single, unified architecture.

**Tokenization and action chunking.** Following RT series and OpenVLA (Brohan et al., 2022; Kim et al., 2024), we discretize each control dimension by a 256-bin quantile-based scheme (discretize only 1st–99th percentiles to avoid outliers), and treat the gripper as a seperate token between 0-1. A single-timestep action therefore consists of $D_{\text{act}}=7$ tokens: 3 for translation, 3 for rotation, and 1 for gripper. For action chunking, we arrange tokens from $H$ future timesteps into a fixed layout, yielding a total of $L=H \times D_{\text{act}}$ action positions. Discrete diffusion model excels at generating fixed length sequences. We additionally append a special [MASK] token to the action vocabulary.

Visual inputs comprise a mandatory third-person image and two optional left and right wrist observations (Fig. 2). Each view is encoded by SigLIP and DINOv2 ViTs, whose features are projected into the Llama2 embedding space; the language instruction is tokenized directly by the Llama2. Optional proprioceptive states (*i.e.* end effector positions) are embedded via a small MLP and concatenated.

**Unified transformer and heads.** All tokens (including vision, language and action) pass through the unified transformer. And we only predict logits at the action positions. For action tokens, we use a *bidirectional* attention mask which means no causal constraint and allow each action position to attend to all vision and language tokens. This design enables naturally full cross-modal fusion without bespoke adapters. Hidden states at action positions are projected to 256-way logit local vocabulary via a shared classification head. Vision and language tokens follow standard VLM masking.

Compared to prior two action head designs, this unified backbone retains the language model's representation power, scales seamlessly with model size, and allows parallel decoding. At inference time, our adaptive re-masking (detailed in Sec. 3.5) further refines uncertain tokens, combining the global-context strength of transformers with the iterative denoising spirit of diffusion.

## 3.4 ALGORITHMIC PIPELINE

**Training pipeline.** During training, for each minibatch, firstly we sample a mask ratio $\gamma \in (0, 1]$ from a schedule (*e.g.*, linear or cosine) that emulates diffusion time. Then we replace $\gamma L$ action positions with special token [MASK], and minimize the masked cross-entropy on those action indices following Eq. 5. We adopt hard-label supervision, representing ground truth as one-hot vectors at the masked indices. Vision and language tokens are attended but ignored in loss. The objective is compatible with LM training which helps preserve the pretrained VLM capability, while aligning action generation with discrete diffusion via mask schedules. Because the model predicts a fixed-length action chunk with $L=H \times D$ tokens, all sequences have uniform length and require no [EOS] padding. Joint optimization over the $L$ positions naturally trains the action chunk in a single pass.

**Inference pipeline.** At test time, we initialize all $L$ action positions as [MASK] and perform a small number of parallel refinement rounds. At each round $t$, our model predicts token posteriors for every currently masked position. We then draw a candidate token at each position by multinomial sampling from the predicted logits. Next, we set the mask ratio to $\gamma_t$ according to a preset schedule and use it to determine how many positions remain masked. We rank the masked positions by data-dependent scores (*e.g.* maximum confidence or confidence gap), commit the sampled tokens at the top $(1 - \gamma_t)$ fraction, and keep the remaining $\gamma_t$ fraction masked for the next round. This schedule makes the decoding order *adaptive* to each instance rather than fixed. The reverse-step conditionals follow the masking Bayesian formula in Eq. 4. A lightweight secondary re-masking mechanism further applies threshold and consistency checks to previously demasked tokens to prevent error propagation. Full criteria are detailed in Section 3.5.

## 3.5 ADAPTIVE DECODING MECHANISM AND SECONDARY RE-MASKING

**Adaptive Decoding Mechanism.** As illustrated above, the inference pipeline starts from a fully masked action chunk $\mathbf{a}_1 = \mathrm{M}^L$ with mask ratio $\gamma_1=1$, and then performs $T$ refinement steps with a monotone schedule $\gamma_{t+1} < \gamma_t$. Here, we use cosine schedule. At step $t$, the model yields per-position posteriors $p_\theta(\mathbf{a}_{t-1} \mid \mathbf{a}_t, \mathbf{c})$ instantiating the reverse conditionals in Eq. 4. We score each position $i$ using one of the *adaptive* metrics:

$$s_{t,i} = \max_k p_\theta(k \mid \mathbf{a}_t, \mathbf{c}) \quad \textbf{(Max Confidence)}, \tag{6}$$

$$g_{t,i} = p_\theta(k_{(1)} \mid \cdot) - p_\theta(k_{(2)} \mid \cdot) \quad \textbf{(Confidence Gap)}, \tag{7}$$

with $k_{(1)}, k_{(2)}$ the labels corresponding to the highest and second-highest probabilities. Let $m_{t,i} \in \{s_{t,i}, g_{t,i}\}$. We keep the top $(1 - \gamma_{t+1})L$ positions $\mathcal{K}_t$ and update these positions' tokens via

Figure 3: **Benchmarks and tasks.** We evaluate Discrete Diffusion VLA across three robot settings: *LIBERO* with a Franka Panda arm, *SimplerEnv–Fractal* with a Google Robot, and *SimplerEnv–Bridge* with a WidowX Arm.

tempered Gumbel sampling to encourage exploration:

$$a_{t+1,i} \sim \text{Categorical}\bigg(\text{softmax}\bigg(\frac{\log p_\theta(\cdot \mid \mathbf{a}_t, \mathbf{c}) + \boldsymbol{\varepsilon}}{\tau_t}\bigg)\bigg), \quad i \in \mathcal{K}_t, \tag{8}$$

where $\boldsymbol{\varepsilon}$ has i.i.d. $\text{Gumbel}(0, 1)$ components (equivalently, Gumbel–Max), and $\tau_t$ is a temperature that decays with $\gamma_t$. Positions not in $\mathcal{K}_t$ are set to [MASK], and the process iterates until $\gamma_T{=}0$ or convergence. This instance-wise ranking makes the decoding order *adaptive* rather than fixed.

**Secondary Re-Masking.** Beyond meeting the target ratio $\gamma_{t+1}$, we run two lightweight checks on previously committed tokens to prevent early errors from persisting. Tokens that fail either checks are re-masked, a step we refer to as secondary re-masking.

*(i) Threshold check.* If the current confidence falls below a monotonically-increasing step-dependent threshold $\eta_t^{\text{abs}}$, the token is re-masked:

$$\mathcal{R}_t^{\text{abs}} = \big\{\, i \in \mathcal{K}_t \ : \ s_{t,i} < \eta_t^{\text{abs}} \,\big\}. \tag{9}$$

*(ii) Residual-drop check.* Let $t_i^\star$ denote the first step at which position $i$ was kept, and cache the reference confidence score $s_i^{\text{ref}} \coloneqq s_{t_i^\star, i}$. We compute a confidence residual $\Delta_{t,i} = s_i^{ref} - s_{t,i}$. Tokens with large degradation are re-masked either by a threshold or top-$Q$ rule:

$$\mathcal{R}_t^{\text{drop}} = \big\{\, i \in \mathcal{K}_t \ : \ \Delta_{t,i} > \eta_t^{\text{drop}} \,\big\} \quad \text{or} \quad \mathcal{R}_t^{\text{drop}} = \underset{Q}{\arg\text{top}}\, \Delta_{t,i}. \tag{10}$$

The secondary re-masked set is $\mathcal{R}_t = \mathcal{R}_t^{\text{abs}} \cup \mathcal{R}_t^{\text{drop}}$. For $i \in \mathcal{R}_t$ we set $a_{t+1,i} = \text{M}$ before proceeding to step $t + 1$. These operations maintain alignment with the Bayes reverse kernel (Eq. 4) while improving cross-iteration consistency.

## 4 EXPERIMENTS

### 4.1 BENCHMARKS AND BASELINES

**Benchmarks.** We evaluate our Discrete Diffusion VLA on three different robot settings as shown in Fig. 3: (i) Franka Panda arm in LIBERO (Liu et al., 2023), using the four suites LIBERO-Spatial, LIBERO-Object, LIBERO-Goal, and LIBERO-Long (10 tasks per suite; 500 expert demos per suite). (ii) Google Robot in SimplerEnv–Fractal (Li et al., 2025), which reports *Visual Matching* and *Variant Aggregation* scores across diverse scene variations. (iii) WidowX Robot in SimplerEnv–Bridge, a real-to-sim evaluation aligned with BridgeData-V2 Walke et al. (2023) tasks. Policies receive only RGB images, that is one third-person view and one wrist view for LIBERO, and a single third-person view for SimplerEnv, together with a language instruction and optional end-effector positions. No depth, affordances, or other auxiliary information are used.

**Baselines.** We compare against representative policies spanning autoregressive (AR) token decoders, separate MLP action decoders and continuous diffusion / flow-matching action heads, covering both models trained *from scratch* and models *fine-tuned* from large pretrained bases.

Table 1: **LIBERO task performance results (%).** Each column is a LIBERO task suite; values are averaged over 500 rollouts per suite (10 tasks × 50 episodes). Methods above the horizontal rule (Diffusion Policy, Seer (scratch), MDT) are *trained from scratch*; those below are *fine-tuned* from pretrained bases. Best and second-best are bold and underlined, respectively.

| Model | Libero-Spatial Success (%) | Libero-Object Success (%) | Libero-Goal Success (%) | Libero-Long Success (%) | Average Success (%) |
|---|---|---|---|---|---|
| Diffusion Policy (Chi et al., 2023) | 78.3 | 92.5 | 68.3 | 50.5 | 72.4 |
| MDT (Reuss et al., 2024) | 78.5 | 87.5 | 73.5 | 64.8 | 76.1 |
| Seer (scratch) (Tian et al., 2025) | – | – | – | 78.7 | – |
| OpenVLA (Kim et al., 2024) | 84.7 | 88.4 | 79.2 | 53.7 | 76.5 |
| Octo-Base (Ghosh et al., 2024) | 78.9 | 85.7 | 84.6 | 51.1 | 75.1 |
| Seer (fine-tuned) (Tian et al., 2025) | – | – | – | 87.7 | – |
| Dita / DiT Policy (Hou et al., 2025) | 84.2 | 96.3 | 85.4 | 63.8 | 82.4 |
| TraceVLA (Zheng et al., 2025) | 84.6 | 85.2 | 75.1 | 54.1 | 74.8 |
| SpatialVLA (Qu et al., 2025) | 88.2 | 89.9 | 78.6 | 55.5 | 78.1 |
| $\pi_0$ + FAST (Pertsch et al., 2025) | 96.4 | 96.8 | 88.6 | 60.2 | 85.5 |
| $\pi_0$ (Black et al., 2024) | 96.8 | **98.8** | 95.8 | 85.2 | 94.2 |
| OpenVLA-OFT (Cont-Diffusion) | 96.9 | 98.1 | 96.2 | 91.1 | 95.6 |
| OpenVLA-OFT (Discrete) (Kim et al., 2025b) | 96.2 | 98.2 | 95.6 | **92.0** | 95.5 |
| GR00T-N1 (Bjorck et al., 2025) | 94.4 | 97.6 | 93.0 | 90.6 | 93.9 |
| Discrete Diffusion VLA | **97.2** | 98.6 | **97.4** | **92.0** | **96.3** |

*Discretized action methods.* RT-1-X / RT-2-X (O'Neill et al., 2024), OpenVLA (Kim et al., 2024), Octo-Small / Octo-Base (Ghosh et al., 2024), HPT (Wang et al., 2024), TraceVLA (Zheng et al., 2025), SpatialVLA (Qu et al., 2025), OpenVLA-OFT (Discrete) (Kim et al., 2025b) and $\pi_0$+FAST (Pertsch et al., 2025) represent AR-style or BERT-style generation of discrete action tokens with a unified VLM backbone or with a separate MLP decoder.

*Continuous diffusion / flow-matching methods.* Diffusion Policy (Chi et al., 2023), MDT (Reuss et al., 2024), DiT Policy (Dita) (Hou et al., 2025), RoboVLM (Li et al., 2024b), $\pi_0$ (Black et al., 2024), OpenVLA-OFT (Cont.-Diffusion) (Kim et al., 2025b) and GR00T-N1 (Bjorck et al., 2025) instantiate denoising or flow-matching heads over continuous action trajectories. Seer is reported in both scratch and fine-tuned forms (Tian et al., 2025).

All baselines are evaluated on at least one of the three benchmarks using official metrics: LIBERO success rates (SR); SimplerEnv–Fractal visual matching and variant aggregation SR; and SimplerEnv–Bridge partial and full SR. Unless otherwise noted, numbers are taken from the original papers or reproduced from open-source implementations under same input modality described above.

## 4.2 OVERALL PERFORMANCE COMPARISONS

**Training details.** We fine-tune Discrete Diffusion VLA on each benchmark from the same VLM backbone as OpenVLA (Prismatic–7B), following the respective official protocols. All input images are resized to 224px × 224px. For LIBERO, we train a separate policy per suite using the provided demonstrations, filtering unsuccessful episodes as in Kim et al. (2025b), and report success rates over the official test episodes. For SimplerEnv–Fractal and SimplerEnv–Bridge, we fine-tune on Fractal (Brohan et al., 2022) and BridgeData-V2 (Walke et al., 2023), respectively. Across all settings, Discrete Diffusion VLA uses our unified transformer with discrete action tokens and a fixed action chunk. Chunk sizes are chosen to match the widely used settings for fair comparison: 8 for LIBERO and SimplerEnv–Fractal but 3 for SimplerEnv–Bridge. At inference, the adaptive decoding runs a small, fixed number of refinement rounds (12 by default) with a cosine mask schedule, which has been shown effective for discrete diffusion decoding (Chang et al., 2022b).

**LIBERO results.** Table 1 reports success rates (SR) on four LIBERO suites. Discrete Diffusion VLA attains the best average SR of 96.3%, with per-suite scores of 97.2% (Spatial), 98.6% (Object), 97.4% (Goal), and 92.0% (Long). Our most comparable baseline is OpenVLA-OFT (Discrete), which uses the same action discretization as ours but decodes via parallel decoding rather than discrete diffusion. Discrete Diffusion VLA reaches 96.3% average SR vs 95.4% for OpenVLA-OFT (Discrete), a +0.9 point gain. These gains at matched tokenization indicate that discrete diffusion decoding provides a consistent advantage over parallel decoding. For reference, the pure AR baseline (OpenVLA) averages 76.5%, underscoring the benefit of moving beyond left-to-right decoding. Against methods trained from scratch, Discrete Diffusion VLA surpasses Diffusion Policy and MDT by +23.9 and +20.2 points on average, respectively. We observe consistent advantages across suites.

Table 2: **SimplerEnv evaluation across different policies on Google Robot tasks**. We report the results of all models pretrained with OXE dataset (O'Neill et al., 2024) and then fine-tuned with Fractal dataset (Brohan et al., 2022).

| Model | Visual Matching | | | | Variant Aggregation | | | | #Overall |
|---|---|---|---|---|---|---|---|---|---|
| | Pick Coke | Mv Near | Drawer | Avg. | Pick Coke | Mv Near | Drawer | Avg. | Average |
| RT-1-X (O'Neill et al., 2024) | 56.7% | 31.7% | 59.7% | 53.4% | 49.0% | 32.3% | 29.4% | 39.6% | 46.5% |
| RT-2-X (O'Neill et al., 2024) | 78.7% | **77.9%** | 25.0% | 60.7% | 82.3% | **79.2%** | 35.3% | **64.3%** | 62.5% |
| Octo-Base (Ghosh et al., 2024) | 17.0% | 4.2% | 22.7% | 16.8% | 0.6% | 3.1% | 1.1% | 1.1% | 9.0% |
| OpenVLA (Kim et al., 2024) | 16.3% | 46.2% | 35.6% | 27.7% | 54.5% | 47.7% | 17.7% | 39.8% | 33.8% |
| HPT (Wang et al., 2024) | 56.0% | 60.0% | 24.0% | 46.0% | – | – | – | – | – |
| Moto (Chen et al., 2025) | 74.0% | 60.4% | 43.1% | 59.2% | – | – | – | – | – |
| RoboVLM (Li et al., 2024b) | 77.3% | 61.7% | 43.5% | 63.4% | 75.6% | 60.0% | 10.6% | 51.3% | 57.4% |
| TraceVLA (Zheng et al., 2025) | 28.0% | 53.7% | 57.0% | 42.0% | 60.0% | 56.4% | 31.0% | 45.0% | 43.5% |
| $\pi_0$ (Black et al., 2024) | 72.7% | 65.3% | 38.3% | 58.8% | 75.2% | 63.7% | 25.6% | 54.8% | 56.8% |
| $\pi_0$-FAST (Pertsch et al., 2025) | 75.3% | 67.5% | 42.9% | 61.9% | 77.6% | 68.2% | 31.3% | 59.0% | 60.5% |
| OpenVLA-OFT (Kim et al., 2025b) | 72.3% | 69.6% | 47.2% | 63.0% | 65.3% | 59.0% | 12.2% | 45.5% | 54.3% |
| GR00T-N1 (Bjorck et al., 2025) | 47.0% | 70.0% | 18.1% | 45.0% | 78.8% | 62.5% | 13.2% | 51.5% | 48.4% |
| Discrete Diffusion VLA | **85.4%** | 67.5% | **60.6%** | **71.2%** | 82.5% | 64.6% | 23.6% | 56.9% | **64.1%** |

Table 3: **SimplerEnv evaluation across different policies on WidowX Robot tasks**. We report the results of all models pretrained with OXE dataset (O'Neill et al., 2024) and then fine-tuned with BridgeData V2 (Walke et al., 2023).

| Method | Put Spoon on Towel | | Put Carrot on Plate | | Stack Green on Yellow | | Put Eggplant in Basket | | #Overall |
|---|---|---|---|---|---|---|---|---|---|
| | Grasp Spoon | Success | Grasp Carrot | Success | Grasp G Block | Success | Grasp Eggplant | Success | Average |
| RT-1-X (O'Neill et al., 2024) | 16.7% | 0.0% | 20.8% | 4.2% | 8.3% | 0.0% | 0.0% | 0.0% | 6.3% |
| Octo-Base (Ghosh et al., 2024) | 34.7% | 12.5% | 52.8% | 8.3% | 31.9% | 0.0% | 66.7% | 43.1% | 31.3% |
| Octo-Small (Ghosh et al., 2024) | 77.8% | 47.2% | 27.8% | 9.7% | 40.3% | 4.2% | 87.5% | 56.9% | 43.9% |
| OpenVLA (Kim et al., 2024) | 4.1% | 0.0% | 33.0% | 0.0% | 12.5% | 0.0% | 8.3% | 4.1% | 7.8% |
| RoboVLM (Li et al., 2024b) | 54.2% | 29.2% | 25.0% | 25.0% | 45.8% | 12.5% | 58.3% | 58.3% | 38.5% |
| $\pi_0$ (Black et al., 2024) | 45.8% | 29.1% | 25.0% | 0.0% | 50.0% | 16.7% | 91.6% | 62.5% | 40.1% |
| $\pi_0$-FAST (Pertsch et al., 2025) | 62.5% | 29.1% | **58.5%** | 21.9% | 54.0% | 10.8% | 83.3% | 66.6% | 48.3% |
| OpenVLA-OFT (Kim et al., 2025b) | 50.0% | 12.5% | 41.7% | 4.2% | **70.8%** | 8.3% | **91.7%** | 37.5% | 39.6% |
| GR00T-N1 Bjorck et al. (2025) | **83.3%** | **62.5%** | 54.2% | **45.8%** | **70.8%** | 16.7% | 41.7% | 20.8% | 49.5% |
| Discrete Diffusion VLA | 70.8% | 29.2% | 58.3% | 29.2% | 62.5% | 20.8% | 91.7% | 70.8% | **54.2%** |

**Google Robot results.** As shown in Tab. 2, Discrete Diffusion VLA achieves the best *Visual Matching* average of 71.2%, clearly surpassing widely used baselines including $\pi_0$ (58.8%) and $\pi_0$-FAST (61.9%), and markedly outperforming OpenVLA-OFT (63.0%). On *Variant Aggregation*, Discrete Diffusion VLA attains 56.9%, competitive with the top RT-2-X (64.3%) and $\pi_0$-FAST (59.0%). Aggregating both metrics, Discrete Diffusion VLA yields the highest overall average of 64.1%, reflecting strong robustness across tasks.

**WidowX Robot results.** On the WidowX evaluation in Tab. 3, Discrete Diffusion VLA attains the best overall average of 54.2%, outperforming diffusion / flow-matching policies ($\pi_0$: 40.1%) and exceeding AR-style and BERT-style baselines (*e.g.*, Octo-Small at 43.9% and $\pi_0$-FAST: 48.3%). Per-task breakdown shows consistent gains in both grasp and success metrics (*e.g.*, Put Eggplant in Basket: 91.7% grasp / 70.8% success), indicating that discrete diffusion decoding improves reliability in these visually diverse manipulation settings.

Across all three settings, the best overall score supports our central claim that casting action generation as discrete diffusion inside a single transformer trained with masked cross-entropy preserves VLM priors while enabling parallel, adaptive, revisitable decoding. This unified, non-autoregressive design consistently outperforms both AR and continuous diffusion / flow-matching baselines under identical action tokenization way.

## 4.3 ABLATION STUDY

**Decoding strategy.** We compare one-shot parallel decoding, random order, confidence-gap selection, max-confidence selection and our max-confidence + secondary remasking. On LIBERO-Goal, the success rates are 95.6%, 96.0%, 96.6%, 97.0%, and 97.4% respectively (Tab. 4). The adaptive easy-first schedule guided by per-token confidence yields $> +1\%$ over one-shot parallel. And max-confidence selection is slightly better than confidence-gap, while random order lags behind. Additionally, our secondary remasking scheme also brings about $+0.4\%$ benefits. These results

Table 4: **Ablation study on decoding strategy.** (LIBERO-Goal). Ranking tokens by instance-wise confidence improves over one-shot parallel by $> 1$ point, and our max conf+secondary remask yields the best accuracy (97.4%).

| Decoding Strategy | Parallel Decoding | +Random Order | +Confidence Gap | +Max Confidence | +Max Conf. +Second Remask |
|---|---|---|---|---|---|
| **Success** | 95.6% | 95.8% | 96.6% | 97.0% | **97.4%** |

Table 5: **Ablation study on choice temperature.** (LIBERO-Goal). Linear decay temperature attains the best 97.4%, encouraging mild exploration early and sharper commitment later.

| Choice Temperature | Hard Sample (Temp=0) | Fixed Temp (Temp=1) | Linear Decay Temp (Temp=1-$t$) |
|---|---|---|---|
| **Success Rates** | 96.2% | 96.4% | **97.4%** |

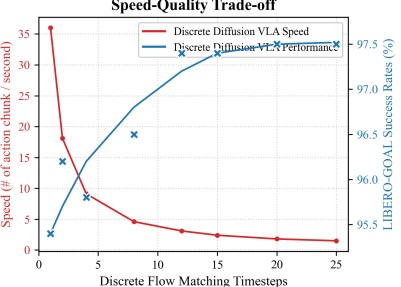

Figure 4: **Speed–Quality trade-off.** (i) Time efficiency by the number of generated action chunks per second. (ii) Ablation on denoising steps. We adopt $T = 12$ as a knee point for high accuracy at strong throughput.

indicate that ranking tokens by instance-wise confidence and resolving the easy elements first improves refinement effectiveness and final accuracy.

**Choice temperature.** We study the temperature used to turn posteriors into discrete choices during refinement. Hard argmax at all steps reaches 96.2%, a fixed temperature of 1.0 gives 96.4%, and a simple linear decay from 1.0 to 0.0 across steps achieves 97.4% (Tab. 5). Decay encourages mild exploration early and sharper commitment later, complementing adaptive decoding to correct early mistakes and consolidate consistent actions.

**Denoise Steps.** Please refer to Fig. 4. And detailed illustrations are in the following section.

### 4.4 ANALYSIS ON INFERENCE EFFICIENCY

Another benefit of discrete diffusion is inference efficiency in number of function evaluations (NFEs) compared to autoregressive decoding. For an action chunk of length $L=H \times D_{\text{act}}$, an AR decoder without mechanisms like multi-token prediction (Gloeckle et al., 2024) performs $L$ sequential forward passes (each token conditions on all previous ones). For instance, in LIBERO, $H=8$ and $D_{\text{act}}=7$ yield $L=56$, creating a latency bottleneck that scales linearly with horizon. It's a critical constraint for real-time robotic applications.

In contrast, Discrete Diffusion VLA denoises the entire chunk in $T$ refinement steps. Each step is a single forward pass that predicts posteriors for all currently masked tokens, so NFEs equal $T$. With our default $T=12$, NFEs drop from 56 to 12 ($4.7\times$ fewer), decoupling cost from sequence length. The adaptive decoding and secondary remasking operate on logits and add no extra forward passes, preserving this efficiency of the parallel refinement.

**Empirical speed measurements.** To validate these theoretical advantages, we measure end-to-end inference latency on a single NVIDIA H800 GPU. Table 6 reports latency and throughput across different methods.

Our method achieves 68.8 ms latency per action chunk with the 7B backbone, $2\times$ faster than AR (136.2 ms) and comparable to optimized continuous diffusion (67.1 ms). While one-shot parallel decoding remains fastest (31.1 ms), it sacrifices performance (e.g., 54.3% vs. 64.1% on SimplerEnv-Fractal). With preliminary KV caching technique on vision-language tokens (Wu et al., 2025), latency reduces to 46.4 ms, a 48% speedup demonstrating promising optimization directions while preserving our iterative refinement advantages. Secondary remasking operates purely on logits with negligible overhead ($<1$ ms), confirming its efficiency.

**Speed–quality trade-off.** Figure 4 sweeps $T$ and reports throughput (number of action chunks generated per second; left $y$-axis) and success rates on LIBERO-GOAL (right $y$-axis). Accuracy improves monotonically with $T$ but with diminishing returns: most gains appear by $T \in [8, 12]$, while increasing to $T \geq 16$ yields sub–1 point improvements at a disproportionate drop in speed (throughput scales roughly as $1/T$). We choose $T=12$ as the optimal operating point balancing accuracy and efficiency.

Table 6: **Inference speed comparison on LIBERO-Goal.**

| Method | Latency (ms) | Speed (Hz) | NFE |
|---|---|---|---|
| OpenVLA (AR) | 136.2 | 7.34 | 56 |
| OpenVLA w/o KVcache (AR) | 209.5 | 4.77 | 56 |
| OpenVLA-OFT (Parallel Decoding) | 31.1 | 32.14 | 1 |
| OpenVLA-OFT (Cont-Diffusion, 50 steps) | 199.9 | 5.00 | 50 |
| OpenVLA-OFT (Cont-Diffusion+, 12 steps) | 67.1 | 14.91 | 12 |
| **Discrete Diffusion VLA (12 steps)** | **68.8** | **14.53** | **12** |
| Discrete Diffusion VLA+ (w/ KV cache) | 46.4 | 21.54 | 12 |

## 5 CONCLUSION

Discrete Diffusion VLA unifies vision, language, and actions inside a single transformer by marrying diffusion's progressive-refinement paradigm with a discrete-token action interface, enabling an adaptive "easy-first, hard-later" decoding order and secondary re-masking for reliable error correction. Our architecture preserves pretrained VLM priors, breaks the left-to-right bottleneck in autoregressive VLA, and delivers state-of-the-art performance across LIBERO and two SimplerEnv suites while using fewer function evaluations than AR baselines. By aligning action decoding with the VLM transformer, Discrete Diffusion VLA offers a path to inherit unified-transformer scaling behavior, paving the way for large-scale VLA research with larger models and broader datasets.

ETHICS STATEMENT

This research adheres to the ICLR 2026 ethical guidelines and upholds the principles of responsible research. We ensure that no personally identifiable, sensitive, or harmful data were used. Our experiments were based on publicly available datasets and did not involve any human subjects or vulnerable groups. We have considered the potential social impact of our methods, including the risk of misuse, and believe that these contributions primarily advance scientific understanding and do not pose foreseeable harm.

REPRODUCIBILITY STATEMENT

We follow the reproducibility guidelines in the ICLR 2026 author guidelines. We will open source code, configuration files, and scripts to reproduce our results, on open platforms such as GitHub.

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

## A DISCLOSURE OF LARGE LANGUAGE MODEL USAGE

We employed large language models (LLMs) only for minor copyediting—grammar and phrasing. All LLM-suggested edits were carefully reviewed and verified by the authors to prevent fabricated content and preserve the original intent. The research ideas, methodology, implementation, experiments, data analysis, and conclusions presented in this work were entirely conceived and executed entirely by the authors without LLM assistance.

## B VISUALIZATIONS OF ROBOT TASK EXECUTIONS

i) Franka Panda Arm on LIBERO-Spatial Task

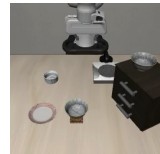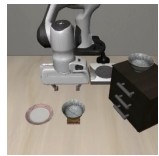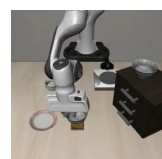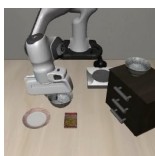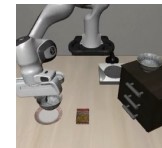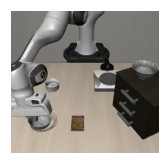

ii) Google Robot on Move Near Task

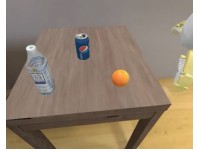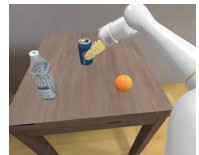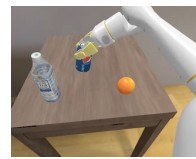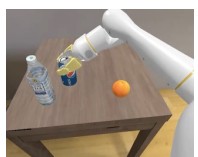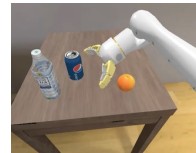

iii) WidowX Arm on Put Eggplant in Basket Task

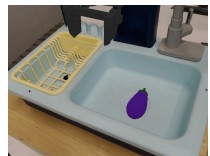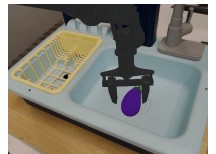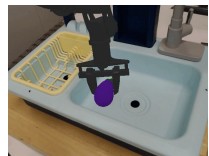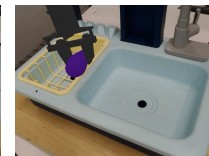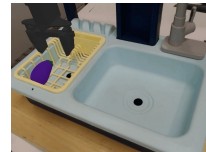

iv) Franka Panda Arm on LIBERO-Long Task

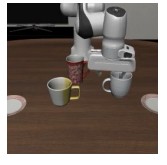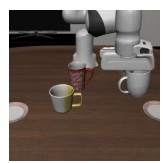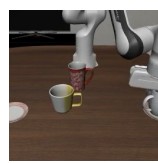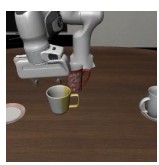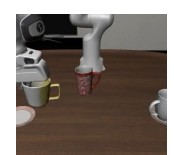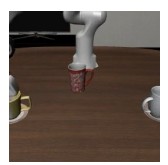

v) WidowX Arm on Stack Green Block on Yellow Block Task

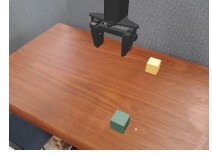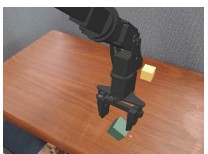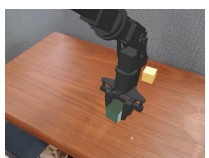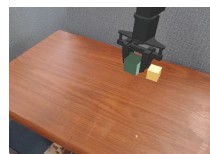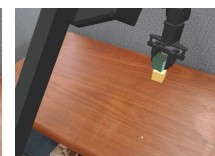

## C IMPLEMENTATION DETAILS

1. We choose action chunk size $H$ as $8$ for both LIBERO and SimplerEnv-Fractal, and $3$ for SimplerEnv-Bridge, following widely used settings in each environment, respectively.

2. We conduct our experiments with batch size 32 for all of the experiments and typically conduct each run on 4 NVIDIA A800 TENSOR CORE GPUs.

3. We apply Feature-wise Linear Modulation (FiLM) (Kim et al., 2025b) on Simpler-Bridge experiments to enhance the language grounding abilities of our model on WidowX Arm manipulation tasks.

4. We train our model on LIBERO-Spatial and LIBERO-Object for 150k steps while 300k steps on LIBERO-Goal and LIBERO-Long. And we report the results of each highest checkpoints. Besides, we only train 100k steps on both Simpler-Fractal and Simpler-Bridge and report the highest overall performance of each environment.

5. For secondary re-masking, we set $\eta_t^{abs} = 0.5 \times (1 - t/T)$ as the step-dependent threshold of threshold check.

## D OUT-OF DISTRIBUTION EVALUATION BENCHMARK

We follow the general experimental settings of LIBERO-PRO (Zhou et al., 2025) while correcting several inconsistencies identified in the currently published implementation.

### D.1 LIBERO-GOAL-OOD

The out-of-distribution (OOD) setting introduces visual perturbations by replacing objects with variants that differ in scale, material properties, and visual appearance. Specifically, the modified scenes include a larger bowl and a stove with metallic luster reflections.

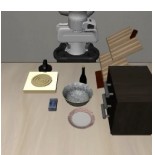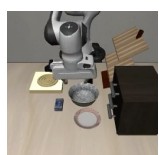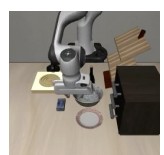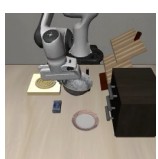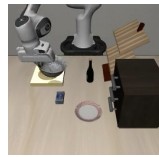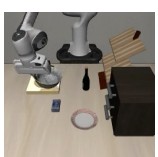

Another case with a bottle and a cabinet constructed from different materials and textures to further challenge the model's generalization capabilities.

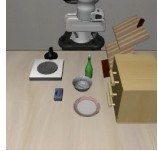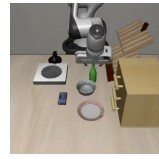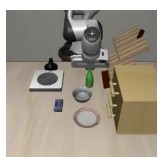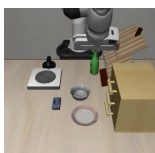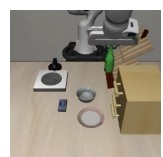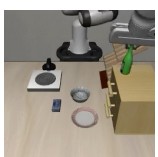

Examples of augmented instructions:

- slice open the top drawer and place the bowl in it
- open the middle dresser of the cabinet
- place the bowl on top of the cabinet

### D.2 LIBERO-SPATIAL-OOD

The visual out-of-distribution setting is similar to that of LIBERO-GOAL-OOD with different scale, material property, and visual appearance objects.

Examples of augmented instructions:

- lift the black bowl on the stove and set it on the plate

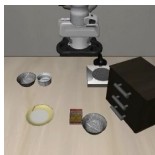 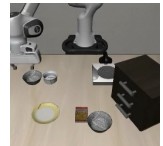 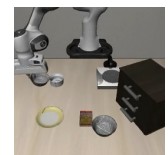 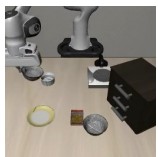 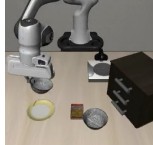 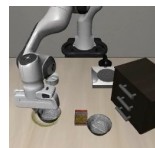

- take the black bowl beside the plate and put it on the plate
- grab the black bowl on the ramekin and set it on the plate

## E   VISUALIZATION AND ANALYSIS OF ADAPTIVE DECODING ORDER

To validate the intuition of Discrete Diffusion VLA's "easy-first, hard-later" decoding pattern, we conducted a comprehensive analysis of the learned adaptive decoding order. We visualized the decoding process for an action chunk across 12 denoising steps, as shown in Figure 5 (a). Each grid represents an $8 \times 7$ layout, where rows correspond to 8 actions in a chunk and columns represent 7-dimensional tokens per action.

We additionally computed the frequency distribution of action tokens in the fine-tuning dataset. Table 5 (b) shows the top 5 most frequent token IDs for each action dimension, with their occurrence percentages. Then in Fig. 5 (a), light gray indicates masked tokens, while varying shades of brown / orange represent demasked tokens, with color intensity encoding token frequency in the training dataset. (Darker colors mean high frequency.)

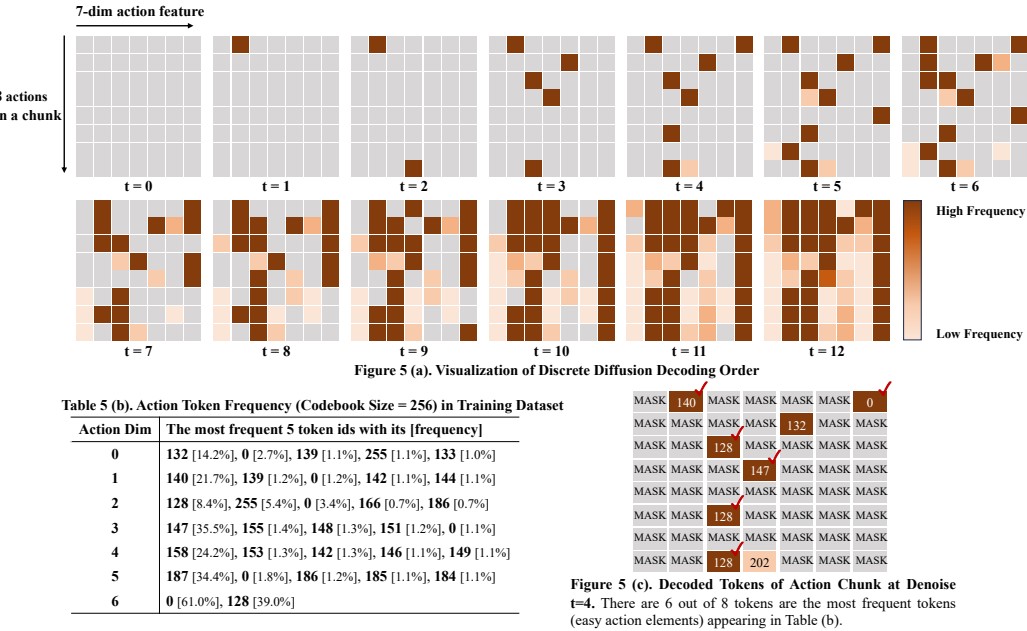

**Figure 5 (a). Visualization of Discrete Diffusion Decoding Order**

**Table 5 (b). Action Token Frequency (Codebook Size = 256) in Training Dataset**

| Action Dim | The most frequent 5 token ids with its [frequency] |
|---|---|
| 0 | **132** [14.2%], **0** [2.7%], **139** [1.1%], **255** [1.1%], **133** [1.0%] |
| 1 | **140** [21.7%], **139** [1.2%], **0** [1.2%], **142** [1.1%], **144** [1.1%] |
| 2 | **128** [8.4%], **255** [5.4%], **0** [3.4%], **166** [0.7%], **186** [0.7%] |
| 3 | **147** [35.5%], **155** [1.4%], **148** [1.3%], **151** [1.2%], **0** [1.1%] |
| 4 | **158** [24.2%], **153** [1.3%], **142** [1.3%], **146** [1.1%], **149** [1.1%] |
| 5 | **187** [34.4%], **0** [1.8%], **186** [1.2%], **185** [1.1%], **184** [1.1%] |
| 6 | **0** [61.0%], **128** [39.0%] |

**Figure 5 (c). Decoded Tokens of Action Chunk at Denoise t=4.** There are 6 out of 8 tokens are the most frequent tokens (easy action elements) appearing in Table (b).

Figure 5: **Visualization and Analysis of Adaptive Decoding Order**

From the figure, we can find that darker colors (high-frequency tokens) are predominantly decoded in the early timesteps ($t = 0 \sim t = 6$). While token frequency does not directly equal confidence, it can serve as a reasonable proxy for easy / hard action elements because tokens that appear more frequently in training tend to be learned more robustly by the model. This generally supports our claimed intuition that the adaptive decoding mechanism tends to resolve easier action elements before harder ones, though the correspondence is not absolute.

Besides, while high-frequency tokens show a tendency to be decoded earlier, the pattern still follows the actual trajectory structure that not all dark tokens are decoded first. This indicates our strategy adapts to trajectory-specific confidence rather than blindly following dataset statistics, avoiding collapse to dataset bias.

Figure 5 (c) further shows the specific decoded tokens at $t = 4$. Among the 8 decoded tokens, 6 out of 8 come from the top-5 most frequent tokens in Table 5 (b), providing quantitative evidence that our adaptive strategy often prioritizes easy action elements in early refinement steps.

# F   RETENTION OF PRETRAINED VISION-LANGUAGE PRIORS: OUT-OF-DISTRIBUTION ANALYSIS

To validate our claim that discrete diffusion VLA preserves pretrained vision-language priors, we evaluate retained capabilities under out-of-distribution perturbations following LIBERO-PRO (Zhou et al., 2025).

**Experimental Setup.** We conduct two OOD tests: (1) **Language Augmentation**, paraphrasing task instructions with diverse linguistic variations (e.g., "open the top drawer and put the bowl inside" → "slice open the top drawer and place the bowl in it"), and (2) **Vision Augmentation**, altering object appearances including materials, colors, and sizes (e.g., bowls scaled by $\sim 1.5\times$). We corrected two bugs in the original LIBERO-PRO implementation related to BDDL file naming and language expansion. Sample augmentations are provided in Appendix D.

**Results.** Tables 7 and 8 present results on LIBERO-Goal and LIBERO-Spatial, respectively.

Table 7: **Out-of-distribution performance on LIBERO-Goal**

| Method | Original | Lang Aug | Vision Aug |
|---|---|---|---|
| OpenVLA-OFT (Discrete) | 95.6% | 87.6% (↓8.0%) | 73.0% (↓22.6%) |
| OpenVLA-OFT (Cont-Diffusion) | 96.0% | 93.6% (↓2.4%) | 67.0% (↓29.0%) |
| **Discrete Diffusion VLA** | **97.4%** | **96.0% (↓1.4%)** | **76.4% (↓21.0%)** |

Table 8: **Out-of-distribution performance on LIBERO-Spatial**

| Method | Original | Lang Aug | Vision Aug |
|---|---|---|---|
| OpenVLA-OFT (Discrete) | 96.2% | 94.6% (↓1.6%) | 95.0% (↓1.2%) |
| OpenVLA-OFT (Cont-Diffusion) | 96.9% | 95.2% (↓1.7%) | 94.6% (↓2.3%) |
| **Discrete Diffusion VLA** | **97.2%** | **96.0% (↓1.2%)** | **96.2% (↓1.0%)** |

**Analysis.** Our method exhibits substantially smaller degradation under both language and vision perturbations. On LIBERO-Goal, language degradation is only 1.4% vs. 8.0% (parallel decoding) and 2.4% (continuous diffusion), while vision degradation is 21.0% vs. 22.6% and 29.0%. On LIBERO-Spatial, degradations are minimal across all methods ($\leq 3.0\%$), reflecting the suite's inherent layout variations during training. The stark contrast with LIBERO-Goal stems from its limited task diversity—ten sparse tasks with large goal-space gaps that encourage overfitting to specific instances rather than generalizable understanding.

**Architectural Insights.** Parallel decoding shows higher language sensitivity, likely due to PaliGemma's language-centric modality alignment. Continuous diffusion exhibits severe vision degradation (↓29.0% on LIBERO-Goal), consistent with prior findings (Yang et al., 2025b; Liu et al., 2024) that separate diffusion action heads become overly vision-dependent. Our unified discrete diffusion framework mitigates both issues through tighter VLM integration, demonstrating superior preservation of pretrained vision-language capabilities.

