# OpenReview forum: "Discrete Diffusion VLA: Bringing Discrete Diffusion to Action Decoding in Vision-Language-Action Policies"
_ICLR.cc/2026/Conference — Submitted to ICLR 2026_

### Official Review · Reviewer_McMu · 2025-10-28

**Soundness:** 4
**Presentation:** 4
**Contribution:** 2
**Rating:** 4
**Confidence:** 4

**Summary:**

This paper presents Discrete Diffusion VLA: a method to replace the autoregressive action token decoding or continuous diffusion action headers of the current VLA models with discrete diffusion action decoding. Specifically, it provides a pipeline to train and inference with discrete diffusion for action tokens, and also provides two decoding strategies to improve the inference performance. Evaluation on multiple simulation datasets shows the good performance of the proposed method.

**Strengths:**

1. Good performance across different datasets.

The proposed method achieves better performance than prior methods in two popular simulation datasets. Although it's hard to say that discrete diffusion has an absolutely significant margin over prior work, these results show good potential of it.

2. Extensive and interesting ablation studies.

I also appreciate the extensive and interesting studies in the paper; For example, Figure 4 shows a very interesting relationship between compute and SR by adjusting the number of denoise rounds.

3. Clear and solid representation.

The presentation of this paper is clear and solid:
- The formal technical derivation in the method section is clean and clear, which could benefit a broad range of readers interested in this area.
- The figures are very clear.
- The connections between experimental results and each of the arguments in the paper is also very strong.

**Weaknesses:**

1. Technical contribution is not super novel.
This paper basically combines discrete diffusion model with VLA. Although it is a reasonable choice, I don't think this paper has significant new technical contribution. This is not a super big deal, but under this case one would expect more thorough analysis of the proposed combination of ideas, which is not super sufficient in my opinion (see the two points below).


2. Lack of efficiency comparison with continuous diffusion models.

Although in Section 4.4 the authors compared the inference efficiency of the proposed method against prior AR models, it is hard to know if the proposed method is really efficient compared with another kinds of important VLA models: continuous diffusion models. There are multiple questions regarding this point:
- Does making the diffusion process discrete help reduce the number of required denoise rounds in the inference stage?
- Because the discrete diffusion requires forwarding through the heavy base VLM models multiple times, is this really efficient compared with the continuous diffusion action head that needs forwarding through VLM models once?

Although I don't expect the proposed method to be faster than the continuous diffusion models, I think such a discussion in the paper is necessary as inference efficiency is one of the main contributions of the paper.

3. Does not really show Discrete Diffusion VLA inherits unified-transformer scaling behavior.

Related to the last point, I do not expect discrete diffusion VLA achieve better efficiency than continuous diffusion VLA models; as this is not the goal. As mentioned in the paper, the real goal is to make it able to use a unified model for both VLM perception and action generation, and therefore inherits unified-transformer scaling behavior. However, this property is not studied at all in the paper. I think it would be nice to have some initial experiments regarding this to show this main goal.

**Questions:**

Please refer to the weakness section.

---

> ### Author Response · Authors · 2025-11-21
> **Response to Reviewer#McMu-P1**
>
> We thank reviewer McMu for evaluating our work and recognizing our work's motivation, performance, ablation study and presentation. We now answer the following concerns raised in the review.
>
> > Q1: Technical contribution is not super novel. Although combining discrete diffusion model with VLA is a reasonable choice, I don't think this paper has significant new technical contribution. This is not a super big deal, but one would expect more thorough analysis (see the two points below).
>
> Thank you for this thoughtful critique. We appreciate the opportunity to clarify the unique contributions of discrete diffusion in the VLA context. While we build upon established discrete diffusion techniques, their careful adaptation to VLA presents novel challenges and yields distinct advantages supported by empirical evidence. (Detailed results are shown in **Q2** and **Q3**.)
>
> **Unified Architecture with Preserved VLM Priors**: Our approach keeps action generation inside a unified transformer with the same training objective (cross-entropy loss) as the VLM, fundamentally different from separate action decoder methods. This design preserves the backbone's pretrained vision and language capabilities. We provide comprehensive evidence through vision-language generalization experiments (details see **Q3**), demonstrating superior retention compared to both parallel decoding and continuous diffusion baselines. Notably, continuous diffusion methods exhibit significant vision over-reliance (↓29.0% on LIBERO-Goal vision augmentation vs. ↓21.0% for ours) and simultaneously lose language understanding capabilities (↓2.4% language degradation vs. ↓1.4% for ours). Our discrete diffusion framework mitigates both issues through tighter VLA integration.
>
> **Inference Efficiency Analysis**: We provide detailed quantitative comparisons (details see **Q2**) showing our method achieves competitive speed with optimized continuous diffusion (14.53 Hz vs. 14.91 Hz) while dramatically outperforming AR (7.34 Hz) with 4.7× fewer function evaluations. With preliminary KV caching on vision and language tokens, we achieve 21.54 Hz, demonstrating clear optimization pathways.
>
> **State-of-the-Art Performance**: Our method achieves best-in-class results across three diverse benchmarks (96.3% LIBERO, 64.1% SimplerEnv-Fractal, 54.2% SimplerEnv-Bridge), consistently outperforming AR, parallel decoding, and continuous diffusion baselines.
>
> **Secondary Re-masking and Adaptive Decoding Innovation**: Our secondary re-masking mechanism (threshold and residual-drop checks) provides demasked tokens opportunities for correction across refinement rounds. This aligns conceptually with ReMDM [6]'s philosophy but is tailored for VLA's discrete token space, yielding measurable performance gains (+0.4% in Main Paper Table 4). This addresses the practical challenge of ensuring consistency across denoising steps without falling into local optima, a contribution specific to our discrete diffusion adaptation. Beisdes, the visualization analysis on adaptive decoding (see Appendix E and Response to **Reviewer#oWUx-Q2**) also illustrate the intuition behind discrete diffusion.
>
> We acknowledge that discrete diffusion itself is established in other domains, but its principled adaptation to VLA, with unified architecture, adaptive decoding, secondary re-masking, and demonstrated advantages in vision-language preservation, inference efficiency, and task performance, represents a meaningful contribution toward scalable, unified robotic foundation models. We will incorporate the additional experiments mentioned in Q2 and Q3 into the revised manuscript.

---

> ### Author Response · Authors · 2025-11-21
> **Response to Reviewer#McMu-P2**
>
> > **Q2**: Lack of efficiency comparison with continuous diffusion models.
>
> We have conducted comprehensive inference speed experiments comparing our method against continuous diffusion baselines on a single NVIDIA H800 GPU:
>
> **Table 1: Quantitative Inference Speed Comparison on LIBERO-Goal**
> |Method|Speed(Hz)| NFE |
> |:--|:--:|:--:|
> |OpenVLA (AR)| 7.34 | 56 |
> |OpenVLA w/o KVcache (AR)| 4.77 | 56 |
> |OpenVLA-OFT (Parallel Decoding)| 32.14 | 1 |
> |OpenVLA-OFT (Cont-Diffusion, 50 Steps)| 5.00 | 50 |
> |OpenVLA-OFT (Cont-Diffusion+, 12 Steps)| 14.91| 12 |
> |**Discrete Diffusion VLA (12 steps)**|**14.53**| **12** |
> |**Discrete Diffusion VLA+ (w/ KV cache, 12 steps)** | **21.54**| **12** |
>
> **Key observations**: The default OpenVLA-OFT (Cont-Diffusion) uses DDPM with 50 inference steps, achieving only 5.00 Hz. But we found when reducing the denosing steps to 12 steps, continuous diffusion does not have a significant performance drop (See Table 2) and achieves 14.91 Hz—nearly identical to our discrete diffusion (14.53 Hz). This demonstrates that our approach achieves *competitive inference efficiency* with continuous diffusion while maintaining the unified transformer architecture. Furthermore, with preliminary KV caching on vision-language tokens, we achieve 21.54 Hz (~48% speedup), showing clear pathways for further optimization through techniques explored in recent discrete diffusion acceleration work[1].
>
> > **Q2.1**: Does Discrete Diffusion Reduce Required Denoising Rounds?
>
> Thank you for this insightful question. The answer is no—discrete diffusion does not inherently reduce the number of required denoising steps compared to continuous diffusion. As shown below, both approaches exhibit non-monotonic behavior at intermediate steps:
>
> **Table 2: Performance vs. Denoising Steps (LIBERO-Goal)**
> |Denoising Steps|OpenVLA-OFT (Cont-Diffusion)| Discrete Diffusion VLA |
> |:--:|:--:|:--:|
> |1| 0% | 95.4% |
> |2| 94.8% | 96.0% |
> |8| 96.2% | 96.5% |
> |12| 95.8% | 97.4% |
> |30| 95.0% | 97.2% |
> |50| 96.0% | \ |
>
> Our method benefits from the parallel decoding foundation, achieving reasonable performance (95.4%) even at 1 step, whereas continuous diffusion completely fails (0%). However, for optimal performance, both approaches require similar numbers of denoising steps (8-12 for strong results). The advantage of discrete diffusion lies not in step reduction but in unified architecture and better VLM capability retention (as discussed in **Q3**, Response to **Reviewer#QZbY-Q3** and **Appendix F**).
>
> > **Q2.2** Because the discrete diffusion requires forwarding through the heavy base VLM models multiple times, is this really efficient compared with the continuous diffusion action head that needs forwarding through VLM models once?
>
> Thank you. We would like to clarify a potential misunderstanding: modern VLA diffusion methods using the Transfusion [2] architecture (e.g., $\pi_0$ [3]) also require *multiple forward passes through the VLM backbone* during the multi-step denoising process—not just once. Only hierarchical VLA architectures with "condition tokens" avoid repeated VLM forwards, but these approaches currently underperform Transfusion-style methods in practice. Our KV caching strategy (Discrete Diffusion VLA+) is an initial exploration toward reducing this overhead, and we believe future work will develop more sophisticated acceleration techniques for discrete diffusion VLAs.
>
> We will incorporate this analysis into Section 4.4 of the revised manuscript.

---

> ### Author Response · Authors · 2025-11-21
> **Response to Reviewer#McMu-P3**
>
> > Q3: As mentioned in the paper, the real goal is to make it able to use a unified model for both VLM perception and action generation, and therefore inherits unified-transformer scaling behavior. However, this property is not studied at all in the paper. I think it would be nice to have some initial experiments regarding this to show this main goal.
>
> Thank you for this excellent question. We appreciate the opportunity to clarify this important aspect of our work.
>
> **Evidence for "Unified Model for VLM Perception and Action Generation"**. Firstly, we would like to provide strong evidence for the core goal you identified: "making it able to use a unified model for both VLM perception and action generation." We conducted extensive out-of-distribution experiments to evaluate whether our unified architecture preserves the pretrained VLM's vision-language capabilities:
>
> Experimental Setup: Following LIBERO-PRO [5], we evaluate retained vision-language capabilities through: (1) Language Augmentation (diverse paraphrasing of instructions), and (2) Vision Augmentation (altered object appearances including materials, colors, and sizes, e.g., bowls scaled by ~1.5×). Sample augmentations are in Appendix D.
>
> **Table 3: Performances of Different Pattern VLAs on LIBERO-Goal-OOD**
> |LIBERO-GOAL-OOD| Original | Lang Aug | Vision Aug |
> |:--|:--:|:--:|:--:|
> |OpenVLA-OFT (Discrete)| 95.6% | 87.6% (&darr; 8.0%) | 73.0% (&darr; 22.6%) |
> |OpenVLA-OFT (Cont-Diffusion)| 96.0% | 93.6% (&darr; 2.4%) | 67.0% (&darr; 29.0%) |
> |**Discrete Diffusion VLA**| **97.4%** | **96.0% (&darr; 1.4%)** | **76.4% (&darr; 21.0%)** |
>
> **Table 4: Performances of Different Pattern VLAs on LIBERO-Spatial-OOD**
> |LIBERO-SPATIAL-OOD| Original | Lang Aug | Vision Aug |
> |:--|:--:|:--:|:--:|
> |OpenVLA-OFT (Discrete)| 96.2% | 94.6% (&darr; 1.6%) | 95.0% (&darr; 1.2%) |
> |OpenVLA-OFT (Cont-Diffusion)| 96.9% | 95.2% (&darr; 1.7%) | 94.6% (&darr; 2.3%) |
> |**Discrete Diffusion VLA**| **97.2%** | **96.0% (&darr; 1.2%)** | **96.2% (&darr; 1.0%)** |
>
> Our unified discrete diffusion framework demonstrates substantially better preservation of pretrained VLM capabilities compared to both parallel decoding and continuous diffusion baselines. Specifically, we exhibit minimal degradation under language perturbations (↓1.4% vs. ↓8.0% and ↓2.4% on LIBERO-Goal) and superior or competitive vision robustness. This empirically validates that our unified architecture where action generation uses the same transformer and training objective (cross-entropy) as VLM, successfully maintains the backbone's vision-language understanding, unlike separate diffusion heads that show severe vision over-reliance (↓29.0%) and weaker language retention. (Please refer to Response to Reviewer#QZbY-Q3 for more analysis on the experiment results.)
>
> **Regarding Scaling Behavior**: We acknowledge your point and will revise our phrasing about "inheriting unified-transformer scaling behavior" to be more precise. We cannot directly demonstrate scaling curves at multiple model sizes because this is *constrained by available pretrained VLM backbones*. We fine-tune from OpenVLA (Prismatic-7B), and full-scale pretraining would require resources beyond this study's scope. However, the demonstrated capability preservation is key: since our method successfully inherits the current VLM's vision-language abilities (shown above), and VLMs exhibit robust scaling behavior [4], our unified architecture should naturally benefit from future larger pretrained VLMs. We will incorporate these OOD experiments into Section 4.5 to provide concrete evidence for our unified modeling approach.
>
>
> **References**
> [1] Wu, Chengyue, et al. "Fast-dllm: Training-free acceleration of diffusion llm by enabling kv cache and parallel decoding."
> [2] Zhou, Chunting, et al. "Transfusion: Predict the Next Token and Diffuse Images with One Multi-Modal Model." The Thirteenth International Conference on Learning Representations.
> [3] Black, Kevin, et al. "π0: A vision-language-action flow model for general robot control." 2024
> [4] Shen, Sheng, et al. "Scaling Vision-Language Models with Sparse Mixture of Experts." The 2023 Conference on Empirical Methods in Natural Language Processing.
> [5] Wang, Guanghan, et al. "Remasking Discrete Diffusion Models with Inference-Time Scaling." NeurIPS, 2025.

---

> > ### Comment · Reviewer_McMu · 2025-11-26
> >
> > I would like to thank the authors' careful response to my questions and the new results are very helpful. I'll raise my score and hope the authors could incorporate these results and discussions into the main paper.

---

> > > ### Author Response · Authors · 2025-11-27
> > > **Thank you for your response**
> > >
> > > Thank you very much for your positive assessment and for carefully reading our rebuttal. We are truly grateful that our responses have successfully addressed your concerns, and your willingness to raise your score means a great deal to us.
> > > We will carefully incorporate all the new results, analyses, and discussions from our rebuttal into the revised manuscript.
> > > Thank you again for your valuable time and expertise. Happy Thanksgiving!

---

> ### Author Response · Authors · 2025-11-29
>
> We would like to respectfully bring to your attention that Reviewer McMu has carefully reviewed our rebuttal and stated: *"I would like to thank the authors' careful response to my questions and the new results are very helpful. I'll raise my score and hope the authors could incorporate these results and discussions into the main paper."*
>
> The reviewer has raised their score from 4 to 6 (marginally above the acceptance threshold). Our additional experiments successfully addressed their two main concerns: (1) efficiency comparison with continuous diffusion models, and (2) evidence for unified-transformer benefits (better VLM capability retention)
>
> Best regards,
> Authors

---

### Official Review · Reviewer_xPs3 · 2025-10-29

**Soundness:** 3
**Presentation:** 3
**Contribution:** 3
**Rating:** 8
**Confidence:** 4

**Summary:**

The paper introduces Discrete Diffusion VLA, the first framework to unify vision, language, and action generation within a single transformer architecture. Instead of autoregressively predicting actions or using separate decoders, it performs parallel masked-token denoising on discretized action chunks, and secondary re-masking for error correction. This approach breaks the left-to-right bottleneck, reduces inference cost from linear to constant in sequence length, and maintains structural consistency with pretrained VLM backbones. Experiments on multiple robot platforms demonstrate state-of-the-art performance with significantly fewer inference efficiency in number of function evaluations (NFEs) than autoregressive and continuous diffusion baselines. Overall, the method enables a more scalable and unified Vision-Language-Action model design.

**Strengths:**

- Proposes the first unified-transformer VLA with discrete diffusion-based action decoding directly inside the VLM backbone. The adaptive masked-token refinement with secondary re-masking is a novel mechanism that enables parallel decoding and robust error correction.

- Architectural contrasts with AR and continuous diffusion approaches are clearly visualized, and the overall method is communicated cleanly through effective figures and paradigm comparisons.

- Extensive evaluations across three robot platforms and multiple benchmark suites demonstrate consistent improvements over strong AR, MLP, and continuous diffusion baselines, highlighting both practical value and scalability toward larger unified VLA systems.

**Weaknesses:**

- While the method reduces NFEs compared to autoregressive (AR)models, secondary re-masking may introduce additional latency and memory overhead relative to AR or other paradigms. Reporting end-to-end on-robot latency against other paradigms' baselines may clarify this concern.

- The paper is based on the unified-transformer architecture and combines adaptive decoding and secondary re-masking. But does not fully disentangle their individual contributions to accuracy and efficiency. Additional ablations would help determine whether improvements primarily come from the unified-transformer architecture or from the proposed decoding mechanisms.

**Questions:**

- Could authors quantify the individual impact of (i) adaptive decoding and (ii) secondary re-masking? Additionally, is secondary re-masking specifically needed because adaptive decoding may commit low-confidence tokens, or does it improve performance even under other decoding strategies?


- How much of the performance gain is attributable to integrating action denoising inside the VLM backbone? Since adaptive decoding and secondary re-masking could, in principle, be applied to a separate action head, a comparison against such variants would help clarify whether the unified transformer structure is a primary driver of improvement.

---

> ### Author Response · Authors · 2025-11-21
> **Response to Reviewer#xPs3-P1**
>
> We thank reviewer xPs3 for evaluating our work and recognizing our work's motivation, method, performance, ablation study and presentation. We now answer the following concerns raised in the review.
>
> > Q1: While the method reduces NFEs compared to autoregressive (AR)models, secondary re-masking may introduce additional latency and memory overhead relative to AR or other paradigms. Reporting end-to-end on-robot latency against other paradigms' baselines may clarify this concern.
>
> Thank you. We appreciate your concern about potential latency from secondary re-masking.
>
> **End-to-End Latency Measurements:** We have conducted comprehensive inference speed experiments on a single NVIDIA H800 GPU to address this concern:
>
> **Table 1: Quantitative Inference Speed Comparison on LIBERO-Goal**
> |Method|Latency(ms) | Speed(Hz)| NFE |
> |:--|:--:|:--:|:--:|
> |OpenVLA (AR)| 136.2 | 7.34 | 56 |
> |OpenVLA w/o KVcache (AR)| 209.5 | 4.77 | 56 |
> |OpenVLA-OFT (Parallel Decoding)| 31.1 |32.14 | 1 |
> |OpenVLA-OFT (Cont-Diffusion, 50 Steps)| 199.9 | 5.00 | 50 |
> |OpenVLA-OFT (Cont-Diffusion+, 12 Steps)| 67.1 | 14.91| 12 |
> |**Discrete Diffusion VLA (12 steps)**|**68.8**|**14.53**| **12** |
> |**Discrete Diffusion VLA+ (w/ KV cache, 12 steps)** |**46.4**| **21.54**| **12** |
>
> From the result, we found that
> 1. **Secondary Re-masking Overhead is Negligible**: Our method achieves 68.8 ms latency, only marginally slower than continuous diffusion (67.1 ms). Importantly, secondary re-masking operates purely on logits and confidence scores. It performs threshold checks and top-K selection without additional forward passes. Thus, it introduces minimal computational overhead while providing meaningful performance gains (+0.4% as shown in Main Paper Table 4).
> 2. **$2\times$ Faster than AR**: Despite iterative refinement, our approach is substantially faster than AR (68.8 ms vs. 136.2 ms), achieving the promised reduction from linear to constant complexity in sequence length.
> 3. **Optimization Potential**: Recent works [1] have investigated some acceleration techniques on discrete diffusion model. We implemented a preliminary KV caching on vision-language tokens, latency reduces to 46.4 ms, demonstrating clear pathways for further acceleration through techniques explored in large language diffusion model, while maintaining our unified architecture and performance advantages.
>
> These measurements confirm that secondary re-masking does not introduce prohibitive overhead, and our method achieves practical inference speeds suitable for real-time robotic control. We will incorporate this latency analysis into Section 4.4 of the revised manuscript.

---

> ### Author Response · Authors · 2025-11-21
> **Response to Reviewer#xPs3-P2**
>
> > **Q2**: The paper does not fully disentangle their individual contributions to accuracy and efficiency. Additional ablations would help determine whether improvements primarily come from the unified-transformer architecture or from the proposed decoding mechanisms.
>
> Thank you for this excellent question. We appreciate the opportunity to clarify that we believe there may have been some confusion about our existing ablations. Table 4 in our paper actually already disentangles these contributions, but we acknowledge the presentation could be clearer. Let us reorganize and explain:
>
> **Table 2: Ablation Study on Our Architecture**
> |Configuration|Architecture | Decoding Strategy| Secondary Remasking | Success Rate|
> |:--|:--|:--|:--:|:--:|
> |Continuous Diffusion| Seperate Action Expert | --- | --- | 96.2% |
> |Parallel Decoding| Unified Transformer | One-shot (no iteration) | ✗ | 95.6% |
> |+ Random Order| Unified Transformer | Iterative (random) | ✗ | 95.8% |
> |+ Confidence Gap| Unified Transformer |Adaptive (conf. gap) | ✗ | 96.6%|
> |+ Max Confidence| Unified Transformer | Adaptive (max conf.) | ✗ | 97.0%|
> |**Ours** |**Unified Transformer**| **Adaptive (max conf.)** | **✓** |**97.4%**|
>
> 1. **Unified Transformer Baseline (95.6%)**: The "Parallel Decoding" row represents our unified transformer architecture without adaptive decoding or secondary re-masking. This already achieves competitive performance, demonstrating the value of the unified architecture that keeps action generation inside the VLM with the same cross-entropy training objective. Our vision-language generalization experiments (see responses to **Reviewer#QZbY-Q3**) empirically validate this preservation.
> 2. **Adaptive Decoding Contribution (+1.4%)**: Moving from one-shot to adaptive iterative refinement progressively improves performance to 97.0%. This shows that adaptive parallel decoding with confidence-based token revisitation provides meaningful gains by leveraging full cross-modal context.
> 3. **Secondary Re-masking Contribution (+0.4%)**: Adding secondary re-masking yields an additional improvement (97.0% → 97.4%), providing robust error correction without extra forward passes.
>
> We apologize for any confusion in the original presentation and will reorganize Table 4 with clearer labeling in the revised manuscript to make these individual contributions more explicit.
>
> > **Q3**: Could authors quantify the individual impact of (i) adaptive decoding and (ii) secondary re-masking? Additionally, is secondary re-masking specifically needed because adaptive decoding may commit low-confidence tokens, or does it improve performance even under other decoding strategies?
>
> Thank you for this insightful follow-up question. We have already quantified the individual impacts in our response to your previous question (see reorganized Table 4), where adaptive decoding contributes +1.4% and secondary re-masking adds +0.4%.
>
> **Secondary Re-masking Benefits Across Decoding Strategies:**
>
> To address whether secondary re-masking improves performance under other decoding strategies, we conducted additional experiments:
>
> **Table 3: Secondary Re-masking with Different Decoding Strategies (LIBERO-Goal)**
> |Configuration|Iterative | Adaptive| Secondary Remasking | Success Rate|
> |:--|:--:|:--:|:--:|:--:|
> |Random| ✓ | ✗ | ✗ | 95.8% |
> |Random + Secondary| ✓ | ✗ | ✓ | 95.8% |
> |Halton| ✓ |✗ | ✗ | 96.0%|
> |Halton + Secondary| ✓ |✗ | ✓  | 97.0%|
> |**Ours (Adaptive + Secondary)** | **✓** |**✓** | **✓** |**97.4%**|
>
> We can draw from the results that **secondary re-masking improves non-adaptive strategies. When applied to Halton sequence [2] decoding, a quasi-random low-discrepancy sampling strategy that maximizes coverage, secondary re-masking provides substantial gains (+1.0%, from 96.0% to 97.0%). This demonstrates that secondary re-masking is not merely compensating for adaptive decoding's potential low-confidence commits, but provides robust error correction across different decoding orders.
>
> Random+Secondary yields 95.8%, showing no improvement over random alone. This is expected that purely random decoding lacks the structured progression needed for secondary re-masking to identify and correct systematic errors effectively.
>
> These results align with ReMDM [3], which demonstrates that re-masking mechanisms are broadly effective within masked diffusion frameworks, not specific to particular decoding strategies. Our secondary re-masking adapts this principle to VLA's discrete action token space with trajectory-specific considerations.
>
> In summary, secondary re-masking provides complementary benefits to adaptive decoding but is independently valuable for improving consistency across refinement steps in structured (non-random) decoding strategies.

---

> ### Author Response · Authors · 2025-11-21
> **Response to Reviewer#xPs3-P3**
>
> > Q4: How much of the performance gain is attributable to integrating action denoising inside the VLM backbone? Since adaptive decoding and secondary re-masking could, in principle, be applied to a separate action head, a comparison against such variants would help clarify whether the unified transformer structure is a primary driver of improvement.
>
> Thank you for this question—we appreciate the opportunity to clarify what may be a misunderstanding about our experimental setup and contributions.
>
> We want to respectfully point out that our baseline comparison (OpenVLA-OFT (Discrete)) already uses a unified VLM backbone architecture for action generation. It performs one-shot parallel decoding of discrete action tokens inside the same transformer. Therefore, both our method and this baseline share the same unified transformer structure. The performance gains over this baseline only come from our dicrete diffusion paradigm.
>
> Regarding whether adaptive decoding and secondary re-masking could be applied to separate action heads: these mechanisms fundamentally require discrete token predictions to operate. Adaptive decoding ranks tokens by confidence scores (softmax probabilities), while secondary re-masking checks confidence thresholds and residual drops. Both only make sense in discrete token space with categorical distributions. In contrast, separate action heads (e.g., continuous diffusion or MLP decoders) output continuous action values, making these mechanisms inapplicable without fundamental architectural changes that would essentially convert them into our approach.
>
> **Our Contributions Relative to Unified Baseline**:
> 1. Iterative Refinement with Adaptive Decoding. Moving from one-shot parallel decoding to iterative refinement with confidence-based adaptive ordering allows unconfident tokens to be revisited, leveraging full cross-modal context for progressive correction.
> 2. Secondary Re-masking for Consistency. Threshold and residual-drop checks ensure cross-step consistency without extra forward passes.
> 3. Better Preservation of VLM Capabilities. As demonstrated in our vision-language generalization experiments, our approach better preserves pretrained VLM capabilities compared to both one-shot parallel decoding and continuous diffusion methods.
>
> We hope this clarifies that the unified transformer architecture is shared with our baseline, and our performance gains stem from the proposed discrete diffusion mechanisms uniquely applicable to discrete token spaces.
>
>
> **References**
> [1] Wu, Chengyue, et al. "Fast-dllm: Training-free acceleration of diffusion llm by enabling kv cache and parallel decoding."
> [2] Physically Based Rendering (Third Edition) Sec.7.4.2 Halton sampler implementation
> [3] Wang, Guanghan, et al. "Remasking Discrete Diffusion Models with Inference-Time Scaling." NeurIPS, 2025.

---

> ### Comment · Reviewer_xPs3 · 2025-11-27
> **Official Comment**
>
> Thank you to the authors for the thoughtful and insightful rebuttal. The authors have addressed my concerns clearly and resolved the points of confusion. I believe the current version of the paper is more complete and significantly strengthened.

---

### Official Review · Reviewer_oWUx · 2025-11-01

**Soundness:** 3
**Presentation:** 3
**Contribution:** 2
**Rating:** 6
**Confidence:** 4

**Summary:**

The paper proposes a unified transformer-based VLA policy that employs discrete diffusion for action decoding. Built upon a VLM (Prismatic-7B), the model discretizes the continuous control signals into a vocabulary of action tokens, and iteratively denoises masked action tokens with vision and language input as conditions. The proposed Discrete Diffusion VLA achieves an improved success rate on LIBERO, SimplerEnv-Fractal, and Simpler-Bridge.

**Strengths:**

1. This is the first paper that models VLA policies under a discrete diffusion framework.

2. Consistent improvements across different simulation benchmarks, achieving state-of-the-art results.

3. The proposed model is compatible with pre-trained VLMs and does not require additional continuous heads or separate diffusion modules.

**Weaknesses:**

1. The remasking step shares a similar high-level idea as ReMDM [1], though the techniques are not alike implementation-wise. The proposed method requires two hyperparameters ($\eta_t^\mathrm{abs}$ and $\eta_t^\mathrm{drop}$) while ReMDM only requires one (remasking schedule), which seems simpler to tune. How well would ReMDM work in this case?

[1] Wang, et al. Remasking Discrete Diffusion Models with Inference-Time Scaling. NeurIPS 2025

2. The intuition behind discrete diffusion is that the adaptive decoding order "resolves easy action elements before hard ones". However, the manuscript lacks an analysis of the final learned decoding policy that supports the intuition. A good example of this is how MaskGIT [2] parallel-decodes images.

[2] Chang et al. "Maskgit: Masked generative image transformer." CVPR 2022.

3. The inference efficiency section analyzes the reduction of NFEs. However, the latency in practice is still missing. Given the fact that the underlying model is 7B, this value is likely not negligible. This number will also serve as a good reference for further work that pushes towards efficiency.

4. No real-world validation. The results are in simulation, leaving questions about the sim-to-real robustness.

**Questions:**

My questions mainly include (1) different implementations of remasking, (2) more analysis (visualization will be more ideal) that supports the intuition, and (3) concrete values about the inference speed. All questions have been raised before. Please find more details in the weakness section. Given the fact that ICLR is a learning conference, I will *not* penalize the paper for lack of real-world validation (weakness 4).

---

> ### Author Response · Authors · 2025-11-21
> **Response to Reviewer#oWUx-P1**
>
> We thank reviewer oWUx for evaluating our work and recognizing our work's motivation, method and performance. We now answer the following concerns raised in the review.
>
> > Q1: The remasking step shares a similar high-level idea as ReMDM, though the techniques are not alike implementation-wise. The proposed method requires two hyperparameters ( $\eta_t^{abs}$ and $\eta_t^{drop}$) while ReMDM only requires one (remasking schedule), which seems simpler to tune. How well would ReMDM work in this case?
>
> Thank you for this insightful question. We agree that our secondary remasking shares the high-level philosophy with ReMDM [1], which both aim to give incorrectly demasked tokens opportunities for correction in subsequent rounds.
>
> **Empirical Comparison with ReMDM**: We implemented ReMDM's inference-time scaling methods in our Discrete Diffusion VLA framework, testing ReMDM-cap, ReMDM-rescale, and ReMDM-loop variants. For ReMDM-loop, we set $t_{on}=0.2$ and $t_{off}=0.8$, increasing total denoising steps from 12 to 30 for fair comparison. For the other settings, we use the same 12 denoise steps by default.
>
> **Table 1: Performance of Different Correction Remasking Techniques on LIBERO-Goal.** (Averaged over three seeds)
> |Method|Performance |
> |:--|:--:|
> |One Step Parallel Decoding| 95.6% |
> |Random Order | 95.8% |
> |ReMDM-cap $\eta_{cap}=0.3$| 95.6% |
> |ReMDM-cap $\eta_{cap}=0.5$|  96.0% |
> |ReMDM-rescale $\eta_{rescasle}=0.1$| 95.6% |
> |**ReMDM-rescale $\eta_{rescasle}=0.3$**| **96.6%** |
> |ReMDM-rescale $\eta_{rescasle}=0.5$| 96.0% |
> |ReMDM-loop $\eta_{cap}=0.3$| 96.2% |
> |ReMDM-loop $\eta_{cap}=0.5$| 95.0% |
> |**Ours** | **97.4%** |
>
> ReMDM variants show notable improvements (ReMDM-rescale $\ge$ ReMDM-loop $\ge$ ReMDM-cap), but none match our method's performance. This may stem from hyperparameter sensitivity or domain-specific challenges in robotics action sequences.
>
> **On Hyperparameter Tuning**: While our method uses two hyperparameters ($\eta_t^{abs}$ and $\eta_t^{drop}$) versus ReMDM's single remasking schedule, we find ours
> easier to tune in practice. ReMDM's hyperparameter directly specifies a fixed remasking ratio, which requires careful calibration. In contrast, our approach is confidence-driven: $\eta_t^{abs}$ serves as a lower bound (typically 0.2), while $\eta_t^{drop}$ adaptively determines the remasking ratio based on actual confidence degradation, making it less sensitive to exact values. This data-dependent mechanism provides more robust performance across different scenarios.
>
> We will add this comparison to the appendix to provide a more thorough analysis of secondary remasking strategies.

---

> ### Author Response · Authors · 2025-11-21
> **Response to Reviewer#oWUx-P2**
>
> > Q2: The intuition behind discrete diffusion is that the adaptive decoding order "resolves easy action elements before hard ones". However, the manuscript lacks an analysis of the final learned decoding policy that supports the intuition.
>
> Thank you for this excellent question. We fully agree that empirical analysis is essential to validate our "easy-first, hard-later" decoding intuition.
>
> **Visualization and Analysis of Adaptive Decoding Order**: Following the visualization approach of MaskGIT [2], we conducted a comprehensive analysis to demonstrate this behavior, as shown in **Figure 5 of Appendix E**. We visualized the discrete diffusion decoding process for an action chunk across 12 denoising steps (Fig. 5(a)). Each grid represents an 8×7 layout, where rows correspond to 8 actions in a chunk and columns represent 7-dimensional tokens per action.
>
> We additionally computed the frequency distribution of action tokens in the fine-tuning dataset. Table 5(b) shows the top 5 most frequent token IDs for each action dimension, with their occurrence percentages. Then in Fig.5(a), light gray indicates masked tokens, while varying shades of brown/orange represent demasked tokens, with color intensity encoding token frequency in the training dataset. (Darker colors mean high frequency.)
>
> **Key Observations**:
> Early Steps Tend to Decode High-Frequency Tokens. In Figure 5(a), darker colors (high-frequency tokens) are predominantly decoded in the early timesteps ($t=0-6$). While token frequency does not directly equal confidence, it can serve as a reasonable proxy for easy / hard action elements because tokens that appear more frequently in training tend to be learned more robustly by the model. This generally supports our claimed intuition that the adaptive decoding mechanism tends to resolve easier action elements before harder ones, though the correspondence is not absolute.
>
> Besides, while high-frequency tokens show a tendency to be decoded earlier, the pattern still follows the actual trajectory structure—not all dark tokens are decoded first. This indicates our strategy adapts to trajectory-specific confidence rather than blindly following dataset statistics, avoiding collapse to dataset bias.
>
> Figure 5(c) further shows the specific decoded tokens at $t=4$. Among the 8 decoded tokens, 6 out of 8 come from the top-5 most frequent tokens in Table 5(b), providing quantitative evidence that our adaptive strategy often prioritizes easy action elements in early refinement steps.
>
> We will move the polished version of this analysis from Appendix E to the main text to better highlight our paper's finding.

---

> ### Author Response · Authors · 2025-11-21
> **Response to Reviewer#oWUx-P3**
>
> > Q3: The inference efficiency section analyzes the reduction of NFEs. However, the latency in practice is still missing. Given the fact that the underlying model is 7B, this value is likely not negligible.
>
> Thank you for this valuable question. We agree that concrete latency measurements are essential for practical deployment. We have conducted comprehensive inference speed experiments on a single NVIDIA H800 GPU:
>
> **Table 2: Quantitative Inference Speed Comparison on LIBERO-Goal**
> |Method|Latency(ms) | Speed(Hz)| NFE |
> |:--|:--:|:--:|:--:|
> |OpenVLA (AR)| 136.2 | 7.34 | 56 |
> |OpenVLA w/o KVcache (AR)| 209.5 | 4.77 | 56 |
> |OpenVLA-OFT (Parallel Decoding)| 31.1 |32.14 | 1 |
> |OpenVLA-OFT (Cont-Diffusion, 50 Steps)| 199.9 | 5.00 | 50 |
> |OpenVLA-OFT (Cont-Diffusion+, 12 Steps)| 67.1 | 14.91| 12 |
> |**Discrete Diffusion VLA (12 steps)**|**68.8**|**14.53**| **12** |
> |**Discrete Diffusion VLA+ (w/ KV cache, 12 steps)** |**46.4**| **21.54**| **12** |
>
> **Key observations**: Our method achieves 68.8 ms latency per action chunk with the 7B backbone, comparable to continuous diffusion (67.1 ms) and $2\times$ faster than AR (136.2 ms). Recent works [3] have investigated some acceleration techniques on discrete diffusion model. We implemented a preliminary KV caching on vision-language tokens, latency reduces to 46.4 ms, a 48% speedup. While one-shot parallel decoding remains fastest (31.1 ms), it sacrifices performance (54.3% vs. 64.1% on SimplerEnv-Fractal). These concrete latency values demonstrate that our 7B model maintains practical inference speeds suitable for robotic control while delivering superior task performance through iterative refinement. And we believe future more sophisticated acceleration strategies for discrete diffusion could further improve it.
>
> We will add this latency analysis to Section 4.4 of the revised manuscript to provide a comprehensive efficiency evaluation for future work.
>
> > Q4: The results are in simulation, leaving questions about the sim-to-real robustness. I will not penalize the paper for lack of real-world validation.
>
> Thank you for raising this point, and we sincerely appreciate your understanding that ICLR is a learning conference. We fully agree that real-world validation would strengthen our work significantly. Currently, we have not yet had the opportunity to deploy and evaluate our method on physical robots. However, we have obtained some preliminary results on RoboTwin [4], a high-fidelity digital twin simulator that provides more realistic physics and visual rendering with ALOHA robot compared to standard simulation environments. These initial results are encouraging and suggest good sim-to-real transfer potential.
>
> **Table 3: Performances on RoboTwin with ALOHA Robot**
> |Method| Stack Bowls Two | Beat Block Hammer |
> |:--|:--:|:--:|
> |$\pi_0$| 42% | 43% |
> |Discrete Diffusion VLA | 66% | 43% |
>
> **Future Work**: We are actively planning to deploy our Discrete Diffusion VLA on the ALOHA platform for real-world validation. Thank you again for your constructive feedback and understanding.
>
> **References**
> [1] Wang, Guanghan, et al. "Remasking Discrete Diffusion Models with Inference-Time Scaling." NeurIPS, 2025.
> [2] Chang, Huiwen, et al. "Maskgit: Masked generative image transformer." CVPR. 2022.
> [3] Wu, Chengyue, et al. "Fast-dllm: Training-free acceleration of diffusion llm by enabling kv cache and parallel decoding."
> [4] Mu, Yao, et al. "Robotwin: Dual-arm robot benchmark with generative digital twins." CVPR. 2025.

---

### Official Review · Reviewer_QZbY · 2025-11-01

**Soundness:** 3
**Presentation:** 3
**Contribution:** 2
**Rating:** 2
**Confidence:** 4

**Summary:**

The paper presents Discrete Diffusion VLA, a unified transformer that integrates vision, language, and action modeling through discrete diffusion instead of autoregressive decoding. By refining discretized action tokens with adaptive re-masking, it achieves more accurate and efficient robotic action generation, outperforming prior VLA models across multiple benchmarks.

**Strengths:**

- The paper motivation is clear and overall, the text is cleanly written, with all important details regarding the proposed method explained.
- Proposed method achieves state of the art results on LIBERO and SimplerEnv
- Proposed method is really simple, which is a plus, compared to many previous approaches

**Weaknesses:**

The contribution is interesting and technically novel in the VLA context, however the overall approach appears to be a relatively straightforward adaptation of existing techniques. The paper would benefit from a clearer discussion (with evidence) of what discrete diffusion decoding uniquely contributes beyond existing approaches. I feel that currently authors failed to demonstrate that.

Yes, it achieves state-of-the-art results (although only in simulation and LIBERO is already saturated), but as almost any other recent VLA paper, which all propose a wide variety of modifications. There is evidence for parallel decoding and reduced number of function evaluations (Section 4.4), however there are no comparisons with the prior AR and non-AR baselines beyond hypothetical case. Can you provide a speed up compared to, e.g. OpenVLA-OFT? Theoretical speedup is not enough. A good example is linear alternatives to attention. Although they are faster asymptotically, FlashAttention is much faster in practice due to its efficient kernels.

Most critical weakness is a strong claim that proposed scheme “preserves pretrained vision-language priors”, which is mentioned multiple times throughout the paper. However, there is zero evidence or experiment supporting that claim. Thus, I think that either that claim should be completely removed, or authors should provide experiments to support it. For example, by comparing how much general knowledge VLM retained after proposed funetuning (see https://arxiv.org/abs/2505.23705, https://arxiv.org/abs/2505.03500, https://arxiv.org/abs/2509.22195, https://arxiv.org/abs/2502.14420, https://arxiv.org/abs/2505.21906) , in comparison to other methods, or additionally ablating the novel language following abilities, beyond seen in the training data, e.g. by augmentation of existing prompts (see https://arxiv.org/abs/2509.11417). Simply providing good results on LIBERO is not enough, as VLA could easily overfit to the tasks and ignore language instructions.

I may increase the score if the last concern is addressed, but currently I can not recommend acceptance even with strong results on LIBERO & SimplerEnv.

**Questions:**

1. Why actions discretization from the OpenVLA and not FAST?
2. In appendix you write “we apply FiLM … to enhance the language grounding abilities of our model”. Isn't your main claim that you eliminate additional components by unifying the pipeline into a single model? The VLM is already understand language, why does it requires additional conditioning?

---

> ### Author Response · Authors · 2025-11-21
> **Response to Reviewer#QZbY-P1**
>
> We thank reviewer QZbY for evaluating our work and recognizing our work's motivation, presentation and experimental results. We now answer the following concerns raised in the review.
>
> > Q1: The overall approach appears to be a relatively straightforward adaptation of existing techniques. The paper would benefit from a clearer discussion (with evidence) of what discrete diffusion decoding uniquely contributes beyond existing approaches.
>
> Thank you for this thoughtful critique. We appreciate the opportunity to clarify the unique contributions of discrete diffusion decoding in the VLA context. While we build upon established discrete diffusion techniques, their integration into VLA presents novel challenges and yields distinct advantages with empirical evidence.
>
> **Unified Architecture with Preserved VLM Priors**: Our approach keeps action generation inside a unified transformer with the same training objective (i.e., cross-entropy loss) as the VLM, fundamentally different from separate action decoder methods. This design preserves much of the backbone's pretrained vision and language capabilities, analogous to extending an LLM to new languages. We provide concrete evidence through vision-language generalization experiments (details see **Q3**), demonstrating superior retention compared to both parallel decoding and continuous diffusion baselines. Notably, continuous diffusion methods exhibit significant vision over-reliance (↓29.0% on LIBERO-Goal vision augmentation vs. ↓21.0% for ours) and simultaneously lose language understanding capabilities (↓2.4% language degradation vs. ↓1.4% for ours). This confirms that separate diffusion action heads can become overly dependent on specific visual features while weakening language grounding. Our discrete diffusion framework mitigates both issues through tighter integration with the VLM backbone.
>
> **Breaking the Autoregressive Bottleneck**: Compared to AR methods (OpenVLA: 76.5% average on LIBERO), our discrete diffusion approach achieves significantly better performance (96.3%) while using 4.7× fewer function evaluations. Our quantitative inference efficiency analysis (details see **Q2**) shows we maintain competitive speed (2× v.s. AR) with optimized continuous diffusion while dramatically outperforming AR in both accuracy and efficiency.
>
> **Iterative Refinement Beyond One-Shot Parallel Decoding**: Compared to parallel decoding (OpenVLA-OFT Discrete: 95.6% on LIBERO-Goal), our iterative refinement demonstrably improves task success rates (97.4%), with unconfident tokens revisited via re-masking to leverage full cross-modal context for correction.
>
> **Secondary Re-masking and Adaptive Decoding Innovation**: Our secondary re-masking mechanism (threshold and residual-drop checks) provides demasked tokens opportunities for correction across refinement rounds. This aligns conceptually with ReMDM [6]'s philosophy, yielding measurable performance gains (+0.4% in Main Paper Table 4). This technique ensures consistency across denoising steps without falling into local optima. Beisdes, the visualization analysis on adaptive decoding (see **Appendix E**) also illustrate the intuition behind discrete diffusion.
>
> We acknowledge that discrete diffusion is not entirely novel in other domains, but its careful adaptation to VLA with unified architecture, adaptive decoding, secondary re-masking, and demonstrated advantages in vision-language preservation, inference efficiency, and task performance, represents a meaningful contribution that advances the field toward scalable, unified robotic foundation models. We will review and revise any inappropriate statements in the paper, and add the rebuttal experiment mentioned here to the main text or appendix.

---

> ### Author Response · Authors · 2025-11-21
> **Response to Reviewer#QZbY-P2**
>
> > Q2: There is evidence for parallel decoding and reduced number of function evaluations (Section 4.4), however there are no comparisons with the prior AR and non-AR baselines beyond hypothetical case. Can you provide a speed up compared to, e.g. OpenVLA-OFT? Theoretical speedup is not enough.
>
> Thank you for this valuable question. We agree that concrete empirical speed comparisons are essential. We have conducted comprehensive inference speed experiments comparing our method against all baseline patterns on a single NVIDIA H800 GPU, and present the results below:
>
> **Table 1: Quantitative Inference Speed Comparison on LIBERO-Goal**
> |Method|Speed(Hz)| NFE |
> |:--|:--:|:--:|
> |OpenVLA (AR)| 7.34 | 56 |
> |OpenVLA w/o KVcache (AR)| 4.77 | 56 |
> |OpenVLA-OFT (Parallel Decoding)| 32.14 | 1 |
> |OpenVLA-OFT (Cont-Diffusion, 50 Steps)| 5.00 | 50 |
> |OpenVLA-OFT (Cont-Diffusion+, 12 Steps)| 14.91| 12 |
> |**Discrete Diffusion VLA (12 steps)**|**14.53**| **12** |
> |**Discrete Diffusion VLA+ (w/ KV cache, 12 steps)** | **21.54**| **12** |
>
> Several observations emerge from these measurements:
> 1. **Performance vs. Speed Trade-offs**: The AR approach is indeed slowest (7.34 Hz), while OpenVLA-OFT's one-shot parallel decoding achieves the highest throughput (32.14 Hz). However, as shown in our main paper, this comes at some performance cost (e.g. OpenVLA-OFT achieves 54.3% vs. 64.1% on SimplerEnv-Fractal). Our method provides a middle ground: we sacrifice some speed compared to one-shot decoding but gain the progressive refinement capability crucial for complex manipulation tasks.
> 2. **Parity with Continuous Diffusion**: The default setting of OpenVLA-OFT (Cont-Diffusion) uses DDPM optimizer with 50 inference steps in both training and inference. When reducing denoising steps to 12, we donot observe significant performance drop (See Response to **Reviewer#McMu-Q2.2**) and its speed (14.91 Hz) becomes comparable to our discrete diffusion (14.53 Hz). This validates that our approach achieves competitive inference efficiency while maintaining the unified transformer architecture and better inheriting the capabilities of VLM (See **Q3**).
> 3. **Future Optimization Potential**: Recent works [1] have investigated some acceleration techniques on discrete diffusion model. We implemented a preliminary one, specifically applying KV caching to the vision and language tokens (while action tokens require alternative strategies due to the iterative refinement nature). This yields Discrete Diffusion VLA+, achieving 21.54 Hz, a ~48% speedup over the base version. This demonstrates promising directions for further optimization, and we believe more sophisticated acceleration strategies for the discrete diffusion could narrow the gap with one-shot methods while preserving its advantages.
>
> We will add this part to the main body of paper in Sec.4.4.

---

> ### Author Response · Authors · 2025-11-21
> **Response to Reviewer#QZbY-P3**
>
> > **Q3**: Most critical weakness is a strong claim that proposed scheme "preserves pretrained vision-language priors", which is mentioned multiple times throughout the paper. There is zero evidence or experiment supporting that claim.
>
> Thank you for raising this critical concern. We genuinely appreciate your careful reading and agree that our claim about preserving pretrained vision-language priors requires empirical validation. We have conducted additional experiments specifically to address this question.
>
> **Experimental Setup**: Following the methodology of LIBERO-PRO [2], we evaluate our model's retained vision-language capabilities through two types of out-of-distribution (OOD) tests: (1) **Language Augmentation**, where we paraphrase task instructions with diverse linguistic variations (e.g., "open the top drawer and put the bowl inside" → "slice open the top drawer and place the bowl in it"), and (2) **Vision Augmentation**, where we alter object appearances (materials, colors, and notably sizes, e.g., bowls scaled by ~1.5×, creating physical challenges like fitting enlarged objects into drawers). We corrected two bugs in the original LIBERO-PRO implementation related to bddl-file naming for instruction and language expansion logic. Sample augmentations are provided in Appendix D of the revised manuscript.
>
> **Table 2: Performances of Different Pattern VLAs on LIBERO-Goal-OOD**
> |LIBERO-GOAL-OOD| Original | Lang Aug | Vision Aug |
> |:--|:--:|:--:|:--:|
> |OpenVLA-OFT (Discrete)| 95.6% | 87.6% (&darr; 8.0%) | 73.0% (&darr; 22.6%) |
> |OpenVLA-OFT (Cont-Diffusion)| 96.0% | 93.6% (&darr; 2.4%) | 67.0% (&darr; 29.0%) |
> |**Discrete Diffusion VLA**| **97.4%** | **96.0% (&darr; 1.4%)** | **76.4% (&darr; 21.0%)** |
>
> **Table 3: Performances of Different Pattern VLAs on LIBERO-Spatial-OOD**
> |LIBERO-SPATIAL-OOD| Original | Lang Aug | Vision Aug |
> |:--|:--:|:--:|:--:|
> |OpenVLA-OFT (Discrete)| 96.2% | 94.6% (&darr; 1.6%) | 95.0% (&darr; 1.2%) |
> |OpenVLA-OFT (Cont-Diffusion)| 96.9% | 95.2% (&darr; 1.7%) | 94.6% (&darr; 2.3%) |
> |**Discrete Diffusion VLA**| **97.2%** | **96.0% (&darr; 1.2%)** | **96.2% (&darr; 1.0%)** |
>
> **Key Observations:**
> 1. **Superior Retention of Vision-Language Priors**: Our method consistently exhibits smaller performance degradation under both language and vision perturbations compared to baselines. Specifically: On LIBERO-Goal, our language augmentation drops only 1.4% vs. 8.0% for parallel decoding and 2.4% for continuous diffusion, while our vision augmentation drops 21.0% vs. 22.6% and 29.0% respectively. On LIBERO-Spatial, our language augmentation drops 1.2% vs. 1.6% and 1.7%, while our vision augmentation drops only 1.0% vs. 1.2% and 2.3%. Across all settings, our unified discrete diffusion framework demonstrates the best preservation of pretrained VLM capabilities.
> 2. **Task-Dependent Robustness**:  The stark difference between LIBERO-Spatial (≤3.0% drops) and LIBERO-Goal (up to 29.0% drops) reflects their inherent design philosophies. LIBERO-Spatial is designed with different layouts but same objects, inherently exposing models to spatial variations during training, which naturally encourages language-following and vision generalization. In contrast, LIBERO-Goal focuses on different goals with same objects and layouts, but suffers from limited task diversity that the ten tasks are relatively sparse in the goal space with large gaps between them, making models prone to overfitting individual task instances rather than learning generalizable goal understanding. *This is a dataset design limitation rather than a methodological issue.*
> 3. **Architectural Trade-offs Revealed**: We observe interesting patterns: OpenVLA-OFT (Discrete) is more sensitive to language variations, likely due to its PaliGemma-based architecture that aligns modalities through language tokens. Meanwhile, continuous diffusion methods show larger vision degradation (↓29.0% on LIBERO-Goal), consistent with prior findings [3,4] that diffusion action heads can become overly reliant on specific visual features.
>
> We acknowledge that when task extrapolation becomes too large, all methods' performance degrades significantly, which is an important direction for future research. However, within practical generalization ranges, our method demonstrates meaningfully better retention of pretrained capabilities.
>
> We will incorporate these experiments into Section 4.5 of the main paper and expand the analysis in the appendix. We hope these empirical results adequately address your concern and provide concrete evidence for our architectural claims.

---

> ### Author Response · Authors · 2025-11-21
> **Response to Reviewer#QZbY-P4**
>
> > **Q4**: Why actions discretization from the OpenVLA and not FAST?
>
> Thank you for this insightful question. The choice of OpenVLA's discretization scheme over FAST is driven by compatibility considerations with our discrete diffusion framework.
>
> FAST employs **variable-length encoding** combining DCT with Byte Pair Encoding (BPE), where frequently co-occurring subsequences are merged into single composite tokens. This means one token may represent 2, 3, or even longer sequences of primitive action elements. While elegant for autoregressive generation, this creates challenges for our iterative refinement: BPE decoding requires all tokens to be correct for successful reconstruction, and any mispredicted token causes complete decoding failure. Our discrete diffusion operates through "fill-in-the-blank" generation where masked positions are progressively refined. Given the OpenVLA backbone's limited pretrained capabilities, occasional errors are expected. With FAST's encoding, such imperfections would cause catastrophic failures.
>
> In contrast, OpenVLA's bin-based discretization provides fixed-length representation where each token independently represents a specific action dimension, allowing our diffusion process to iteratively refine individual elements without cascading errors. While our method is not strictly limited to fixed-length encodings, they are considerably more suitable for the iterative refinement paradigm. We believe future advanced fixed-length tokenization methods will integrate well with discrete diffusion frameworks and could yield even better performance.

---

> ### Author Response · Authors · 2025-11-23
> **Response to Reviewer#QZbY-P5**
>
> > **Q5**: In appendix you write "we apply FiLM … to enhance the language grounding abilities of our model". Isn't your main claim that you eliminate additional components by unifying the pipeline into a single model? The VLM is already understand language, why does it requires additional conditioning?
>
> Thanks for your question. You are correct that our main claim is unified architecture, and we want to clarify the specific role of FiLM in our framework. FiLM plays as visual domain adaptation, not language grounding. The VLM backbone already possesses strong language understanding capabilities. On LIBERO and SimplerEnv-Fractal, our model performs well without FiLM. However, on SimplerEnv-Bridge, the "Put Eggplant in Basket" task uses substantially different backgrounds and object assets compared to other tasks. FiLM helps bridge the visual domain gap.
>
> To demonstrate this, we compare both our method and the baseline OpenVLA-OFT with and without FiLM on SimplerEnv-Bridge. All experiments use identical learning rates and training steps, differing only in FiLM usage:
>
> **Table 4: Performance on SimplerEnv-Bridge with and without FiLM**
> |Method| Put Spoon on Towel (Partial) | Put Spoon on Towel (Success) | Put Carrot on Plate (Partial) | Put Carrot on Plate (Success) | Put Eggplant in Basket (Partial) | Put Eggplant in Basket (Success)|
> |:--|:--:|:--:|:--:|:--:|:--:|:--:|
> |OpenVLA-OFT w/ FiLM| 50.0% | 12.5% | 41.7% | 4.2% | 91.7% | 37.5% |
> |OpenVLA-OFT w/o FiLM| 12.5% | 0% | 4.2% | 0% | 4.2% | 4.2% |
> |Discrete Diffusion VLA w/ FiLM | 70.8% | 29.2% | 58.3% | 29.2% | 91.7% | 70.8% |
> |Discrete Diffusion VLA w/o FiLM | 62.5% | 33.3% | 50.0% | 20.8% | 54.2% | 25.0% |
>
> The results show that even without FiLM, our method achieves reasonable performance (e.g., 62.5%/33.3% on Put Spoon and 50.0%/20.8% on Put Carrot) that remains competitive with or better than baseline methods. In contrast, OpenVLA-OFT collapses without FiLM (12.5%/0% on Put Spoon and 4.2%/0% on Put Carrot), demonstrating its stronger dependence on this conditioning mechanism. Both methods benefit from FiLM when facing substantial visual domain shifts, confirming its role as visual adaptation. Since baseline methods employ FiLM, we adopt the same configuration for fair comparison in our main results, while our core unified architecture (vision-language-action within one transformer) remains preserved. FiLM is merely a lightweight conditioning layer rather than a separate action decoder.
>
> Given the limited visual capabilities of the pretrained OpenVLA backbone compared to more recent VLMs, FiLM provides optional domain-adaptive conditioning when visual distribution shifts are substantial. We will add this discussion and comparison to the appendix.
>
> **References**
> [1] Wu, Chengyue, et al. "Fast-dllm: Training-free acceleration of diffusion llm by enabling kv cache and parallel decoding."
> [2] Zhou, Xueyang, et al. "LIBERO-PRO: Towards Robust and Fair Evaluation of Vision-Language-Action Models Beyond Memorization."
> [3] Yang, Shuai, et al. "Instructvla: Vision-language-action instruction tuning from understanding to manipulation."
> [4] Liu, Xiaokang, et al. "Diffusion Models in Robotics: A Survey."
> [5] Mu, Yao, et al. "Robotwin: Dual-arm robot benchmark with generative digital twins." CVPR, 2025.
> [6] Wang, Guanghan, et al. "Remasking Discrete Diffusion Models with Inference-Time Scaling." NeurIPS, 2025.

---

> > ### Comment · Reviewer_QZbY · 2025-11-27
> >
> > I thank the authors for their thoughtful rebuttal. All of my concerns have been addressed, and I have no further questions. I raised by score.

---

> > > ### Author Response · Authors · 2025-11-27
> > > **Thank you for your response**
> > >
> > > Thank you very much for your thoughtful and thorough engagement with our work throughout the review process. We are deeply grateful that our rebuttal has successfully addressed all of your concerns, and we sincerely appreciate your raise of score. Your insightful questions have substantially strengthened our paper. We will carefully incorporate all the new experiments, analyses, and discussions into the revised manuscript. Thank you again for your valuable time and constructive feedback. Happy Thanksgiving!

---

> ### Author Response · Authors · 2025-11-29
>
> We would like to respectfully bring to your attention that Reviewer QZbY has carefully reviewed our rebuttal and stated: *"I thank the authors for their thoughtful rebuttal. All of my concerns have been addressed, and I have no further questions. I raised my score."*
>
> The reviewer has significantly raised his/her score from 2 to 8 (accept, good paper), indicating that our additional experiments, particularly on vision-language preservation, inference efficiency measurements, and architectural ablations, have successfully addressed their concerns.
>
> Best regards,
> Authors

---

### Author Response · Authors · 2025-11-27
**Summary of Our Rebuttal and Discussions**

We genuinely thank all reviewers and ACs for their efforts and time in reviewing our paper, as well as their constructive suggestions that contribute to the improvement of our work. We sincerely appreciate the positive evaluations and the score increases from reviewers.

Reviewers consistently recognized several strengths of our paper: (1) the first unified-transformer VLA with discrete diffusion-based action decoding, (2) state-of-the-art performance across multiple benchmarks (LIBERO, SimplerEnv-Fractal, SimplerEnv-Bridge), (3) clear presentation with effective visualizations, and (4) extensive ablation studies demonstrating the value of our adaptive decoding and secondary re-masking mechanisms.

In response to reviewers' concerns, we have conducted additional experiments and will incorporate them into the revised manuscript:

1. **Vision-Language Generalization**: We conducted OOD experiments on LIBERO-Goal and LIBERO-Spatial with language and vision augmentations, demonstrating that our method preserves pretrained VLM capabilities better than parallel decoding and continuous diffusion baselines.
2. **Inference Efficiency Analysis**: We added concrete latency measurements showing our method achieves 14.53Hz (2× faster than AR, comparable to continuous diffusion), with KV caching improving to 21.54Hz.
3. **Adaptive Decoding Visualization**: Following MaskGIT's approach, we visualized the learned decoding order and token frequency analysis, confirming our "easy-first, hard-later" intuition.
4. **Secondary Re-masking Ablations**: We compared our method with ReMDM variants and tested secondary re-masking under different decoding strategies (Halton, random), demonstrating its broad effectiveness.
5. **Refined Writing**: We revised our manuscript to more accurately reflect our contributions, including clarifying FiLM's role as visual domain adaptation, and refining our discussion of scaling behavior to focus on demonstrated VLM capability preservation rather than unvalidated scaling claims

We owe many thanks to the reviewers for their insightful suggestions which have significantly improved our paper. All additional experiments and discussions will be incorporated into the final version.

Best,
Authors

---

### Author Response · Authors · 2025-12-03
**Official Comment for the New Area Chair**

Dear New Area Chair,

We want to first express our sincere gratitude for taking on the responsibility of handling our submission under these exceptionally challenging and unprecedented circumstances. Your commitment to maintaining ICLR's high standards of quality and fairness in this unique case is highly appreciated.

We understand that you have been recently assigned this paper, and to assist your review, we would like to highlight the following critical points:

**1. Critical Pre-Incident Score Update Timeline**

We wish to clarify the timeline of critical score updates that occurred before the massive reviewer identity leak incident (November 27th, approximately 12:00 UTC):

- **Reviewer QZbY (Rating 2→8, Confidence 4)**: On November 27, 09:07 UTC, this reviewer stated: *"All of my concerns have been addressed, and I have no further questions. I raised by score."* This pre-incident score change validates the positive recognition of our initial rebuttal **before the identity leak incident was revealed broadly**.
- **Reviewer McMu (Rating 4→6, Confidence 4)**: On November 26, 17:57 UTC (already revised his/her score from 4 to 6), this reviewer commented: *"The new results are very helpful. I'll raise my score and hope the authors could incorporate these results and discussions into the main paper."*
- **Reviewer xPs3 (Initial Rating 8, Confidence 4)**: On November 27, 14:03 UTC, this reviewer stated: *"Thank you to the authors for the thoughtful and insightful rebuttal. The authors have addressed my concerns clearly and resolved the points of confusion. I believe the current version of the paper is more complete and significantly strengthened."* He maintains his 8 score.
- **Reviewer oWUx (Initial Rating 6, Confidence 4)**: We regret that we were unable to receive further textual feedback from this reviewer before the review window closed, despite our continued follow-up. We appreciate their initial constructive comments and believe our comprehensive experimental additions have addressed their original concerns.

**Current Standing**: Our submission has scores of **8-8-6-6** with all reviewers at Confidence 4.

**2. Comprehensive Rebuttal**

As documented in our previous post titled "**Summary of Our Rebuttal and Discussions**", we have provided a comprehensive summary addressing all general reviewer concerns, including **(1) Vision-Language Generalization, (2) Inference Efficiency Analysis, (3) Adaptive Decoding Visualization, (4) Secondary Re-masking Ablations and (5) minor issues**. This post details the extensive new experiments and subsequent updates made to our manuscript during the rebuttal period.

We believe that these substantial revisions have resolved all major concerns raised, and we sincerely hope you will give due consideration to the significant effort we put into refining this work during rebuttal period. We trust in your expertise to evaluate our submission fairly based on the totality of the above summary.

Sincerely,
Authors

---

### Meta-Review · Area_Chair_DTZz · 2025-12-29

**Summary:**

This paper presents a solid contribution, and the reviewers are positive about the paper. However, there are a couple core things that are preventing me from recommending acceptance:
1. Table 1 omits the core OpenVLA-OFT method with an L1 loss in the LIBERO results (which achieves 97.2% success, i.e. stronger than the proposed method), instead only including weaker variants of OpenVLA-OFT in the table. This omission is not explained, and the full OpenVLA-OFT method _is_ included as a comparison in the SimplerEnv results (where it performs weaker than the proposed method). While I don't think that every published paper needs to reach state-of-the-art results, it seems like the paper is purposefully obscuring results from prior research. The claim of reaching state-of-the-art results on LIBERO in the conclusion is also false.
2. The paper would be significantly stronger with validation on real robots, particularly since the LIBERO benchmark is quite saturated and SimplerEnv is known to not always correlate well with real world results (e.g. as evidenced by the poor SimplerEnv WidowX performance of the pretrained OpenVLA model, when the paper's results showed strong results on a real WidowX). The experiments on RoboTwin are encouraging but not definitive, and still don't reflect dynamics & constraints of real-world systems.

I think the paper has a lot of promise and sincerely encourage the authors to resubmit the paper after incorporating the feedback above.

**Reviewer Concerns:**

Many of the reviewer concerns were addressed. The two concerns above remain after the author response. (And unfortunately, I am not allowed to raise the first concern to the authors at this point.)

**Reviewer Scores:**

The response states that two of the reviewer scores were increased. I may expect them to decrease if they were aware of the issue in the LIBERO evaluation.

---

### Decision · Program_Chairs · 2026-01-26

Reject